# Response of vegetation and carbon fluxes to brown lemming herbivory in Northern Alaska

Jessica Plein[1], Rulon W. Clark[1], Kyle A. Arndt[1,2], Walter C. Oechel[1,3], Douglas Stow[4], Donatella Zona[1,5]

[1]Department of Biology, San Diego State University, 5500 Campanile Dr., San Diego, CA 92182, USA
[2]Woodwell Climate Research Center, 149 Woods Hole Rd., Falmouth, MA 02540-1644, USA
[3]Department of Geography, College of Life and Environmental Sciences, University of Exeter, Exeter EX4 4RJ, UK
[4]Department of Geography, San Diego State University, 5500 Campanile Dr., San Diego, CA 92182, USA
[5]School of Biosciences, University of Sheffield, Western Bank, Sheffield S10 2TN, UK

*Correspondence to*: Jessica Plein (jlplein@sdsu.edu, jessicalynnplein@gmail.com)

**Abstract.** The warming of the Arctic is affecting the carbon cycle of tundra ecosystems. Most research on carbon fluxes from Arctic tundra ecosystems has focused on abiotic environmental controls (e.g., temperature, rainfall, or radiation). However, Arctic tundra vegetation, and therefore the carbon balance of these ecosystems, can be substantially impacted by herbivory. In this study we tested how vegetation consumption by brown lemmings (*Lemmus trimucronatus*) can impact carbon exchange of a wet-sedge tundra ecosystem near Utqiaġvik, Alaska during the summer, and the recovery of vegetation during a following summer. We placed brown lemmings in individual enclosure plots and tested the impact of lemmings' herbivory on carbon dioxide ($CO_2$) and methane ($CH_4$) fluxes and the normalized difference vegetation index (NDVI) immediately after lemming removal and during the following growing season. During the first summer of the experiment, lemmings' herbivory reduced plant biomass (as shown by the decrease in the NDVI) and decreased net $CO_2$ uptake, while not significantly impacting $CH_4$ emissions. $CH_4$ emissions were likely not significantly affected due to $CH_4$ being produced deeper in the soil and escaping from the stem bases of the vascular plants. The summer following the lemming treatments, NDVI and net $CO_2$ fluxes returned to magnitudes similar to those observed before the start of the experiment, suggesting a complete recovery of the vegetation, and a transitory nature of the impact of lemming herbivory. Overall, lemming herbivory has short-term but substantial effects on carbon sequestration by vegetation and might contribute to the considerable interannual variability in $CO_2$ fluxes from tundra ecosystems.

## 1 Introduction

The Arctic is warming at about three times the rate of the global average (IPCC, 2021), impacting tundra vegetation and the carbon cycle. Vegetation influences the carbon stored in the tundra ecosystem through the exchange of carbon dioxide ($CO_2$) and methane ($CH_4$) from the soil into the atmosphere via respiration or by $CO_2$ uptake through photosynthesis. One of the largest natural reservoirs of organic carbon in the world is stored within Arctic soils, containing approximately 1,300 Pg of soil organic carbon (Hugelius et al., 2014). Once soils thaw, microbes can convert stored carbon into greenhouse gases that enter the atmosphere, contributing to global warming (McGuire et al., 2009; Schuur et al., 2008). This positive feedback could have dramatic effects on warming rates, and these effects are why most carbon cycle research in tundra systems focuses on abiotic controls on carbon fluxes (Kwon et al., 2019; Oechel et al., 2014; Sturtevant et al., 2012; Zona et al., 2010). Most of the studies investigating the patterns and controls on the carbon balance from Arctic ecosystems focused on the environmental controls on $CO_2$ and $CH_4$, while overlooking the role of herbivory. Since herbivores remove photosynthetic tissues of vegetation, herbivory should substantially decrease the ability of vegetation to photosynthesize and sequester $CO_2$ (Metcalfe and Olofsson, 2015). The decrease in vascular plant cover should also decrease $CH_4$ emissions, given that aerenchyma in sedges (*Carex aquatilis* is the dominant vascular plant and sedge in our study site; Davidson et al., 2016) facilitate the escape of $CH_4$ from deeper anoxic soil layers into the atmosphere

(Dias et al., 2010; McEwing et al., 2015; Ström et al., 2003; Whiting and Chanton, 1993). In addition to transport, vascular plants
also affect the release of labile carbon from photosynthetic tissues, which in turn stimulates $CH_4$ emission (Bhullar et al., 2014;
McEwing et al., 2015; Ström et al., 2003; Tan et al., 2015). Investigating the impacts of herbivory on Arctic vegetation and its
recovery after herbivory would contribute to a refined understanding of the response of tundra ecosystems to climate change.
Small rodents, especially lemmings, in the Arctic tundra of Alaska are important herbivorous consumers of plants and prey
species for larger animals (Le Vaillant et al., 2018). Throughout the Arctic, lemmings are by far the most abundant and widespread
rodent species, and are identified as keystone species in tundra environments (Krebs, 2011). As dominant year-round grazers in
the tundra, lemmings may heavily impact plant productivity (Olofsson et al., 2014). The site of our research, Utqiaġvik, Alaska,
was an ideal site for studying the impact of lemmings on vegetation, as it has been reported that brown lemmings (*Lemmus*
*trimucronatus*) deplete 100 times more primary production than caribou, a much larger herbivorous mammal that migrates
throughout the Alaskan Arctic (Batzli et al., 1980). Due to their life history characteristics and abundance, lemmings can have a
significant influence on the surrounding environment. Lemmings experience cyclic population dynamics where their population
density oscillates, changing community interactions (Soininen et al., 2017). Lemming grazing during population peaks can
dramatically affect vegetation (Olofsson et al., 2012), and therefore greenhouse gas fluxes from Arctic tundra; given the amount
of vegetation consumed by lemmings, their presence could have substantial impacts on the carbon balance of tundra ecosystems.
However, despite the role of lemmings as keystone herbivores, the direct impact of their vegetation consumption on the carbon
cycle of Arctic tundra in Alaska is still largely unknown, with few published studies evaluating the role of lemming herbivory on
the Arctic carbon balance and vegetation (Lara et al., 2017; Lindén et al., 2021; Metcalfe and Olofsson, 2015; Min et al., 2021).
Most of the studies analyzing the effects of lemmings on vegetation focused on ecosystem functioning in the absence of
lemmings (Lara et al., 2017; Lindén et al., 2021; Min et al., 2021), the impacts of lemming waste products and carcasses on nutrient
cycling and vegetation (McKendrick et al., 1980; Roy et al., 2020), the disturbance to soil via turnover by burrowing and fecal
production (McKendrick et al., 1980), and recruitment and loss of forest vegetation (Ericson, 1977). The current body of literature
does not explore the direct impact of lemming presence on carbon cycling and vegetation recovery, leaving a crucial gap in our
understanding of how one of the main herbivores influences this rapidly changing ecosystem, especially in light of Arctic warming.
Since population cycles vary by species and region (Reid et al., 1995), qualitative predictions on how brown lemmings would alter
Arctic vegetation and carbon cycling are uncertain.
In this study, we used enclosures to directly quantify impacts of lemming herbivory on tundra carbon cycling, both immediately
after herbivory and during the following growing season to examine vegetation recovery after one year. Thus far, very few studies
(Johnson et al., 2011; Lara et al., 2017; Lindén et al., 2021; long-term exclosures) have investigated the effect of lemming herbivory
on the tundra carbon cycle, including the timing of recovery of vegetation after lemming herbivory. By using enclosures to
manipulate the number of lemmings per plot and observe a direct impact of lemming presence during peak annual activity, our
study quantified the short-term effects of vegetation removal from lemming herbivory on carbon fluxes and the timing of vegetation
recovery in the Alaskan Arctic.
The short-term effects of brown lemmings' herbivory on Arctic vegetation and carbon fluxes and longer-term recovery are
critical to understand how lemmings might influence tundra environments. For this purpose, we measured the impact of brown
lemmings on vegetation in summer 2018 across a variety of plots in a wet-sedge tundra ecosystem in the Alaskan Arctic. Then, in
summer 2019, we measured vegetation in the plots again to evaluate vegetation recovery from lemmings' grazing. The goal of this
experiment was to understand: (1) how brown lemmings affect vegetation through herbivory and disturbance, and therefore how
they could impact the Arctic tundra carbon cycle and photosynthetic capacity of vegetation, and (2) the rate of vegetation recovery
after brown lemming herbivory.
We hypothesized that lemmings, given their high rate of vegetation consumption, would have a negative impact on net $CO_2$
sequestration by vegetation, but due to the rapid regrowth of the annual vascular plants they preferentially consume, the vegetation
would fully recover in terms of biomass and $CO_2$ sequestration the growing season following grazing. We expected $CH_4$ emission
to decrease in response to herbivory, given the reduction in the biomass of vascular plants. Our broader goals were to increase our
understanding of how the foraging behaviors of these herbivores impacted $CO_2$ and $CH_4$ fluxes and the photosynthetic capacity of
plants in the Alaskan Arctic environment, which we hoped would further public interest in the understanding of complex
interactions in the Arctic and relationships that may exist between climate change, herbivory, and predator-prey interactions.
## 2 Materials and methods
### 2.1 Study location
This study was carried out in Utqiaġvik (formerly Barrow), Alaska (Fig. 1a). Located in the Arctic Coastal Plain, Utqiaġvik is
comprised of polygonal ground (flat-, low-, and high-center ice-wedge polygons) that cover roughly 65 % of the land cover
(Billings and Peterson, 1980). The major vegetation type at this site is graminoid-dominated wetlands, consisting of mosses,
lichens, graminoids (grasses), and wet sedges (Davidson et al., 2016).

91        The study area was located near the Barrow Atmospheric Baseline Observatory, an atmospheric monitoring site managed by

the National Oceanic and Atmospheric Administration (NOAA) (Fig. 1b), approximately 2 km south of the Arctic Ocean
(71°19′21.10″ N: 156°36′33.04″ W). This site was near a pre-established remote flux and meteorological tower monitored by the
Global Change Research Group (Goodrich et al., 2016) and had substantial lemming populations relative to other Arctic tundra
areas in Alaska (Ott and Currier, 2012).

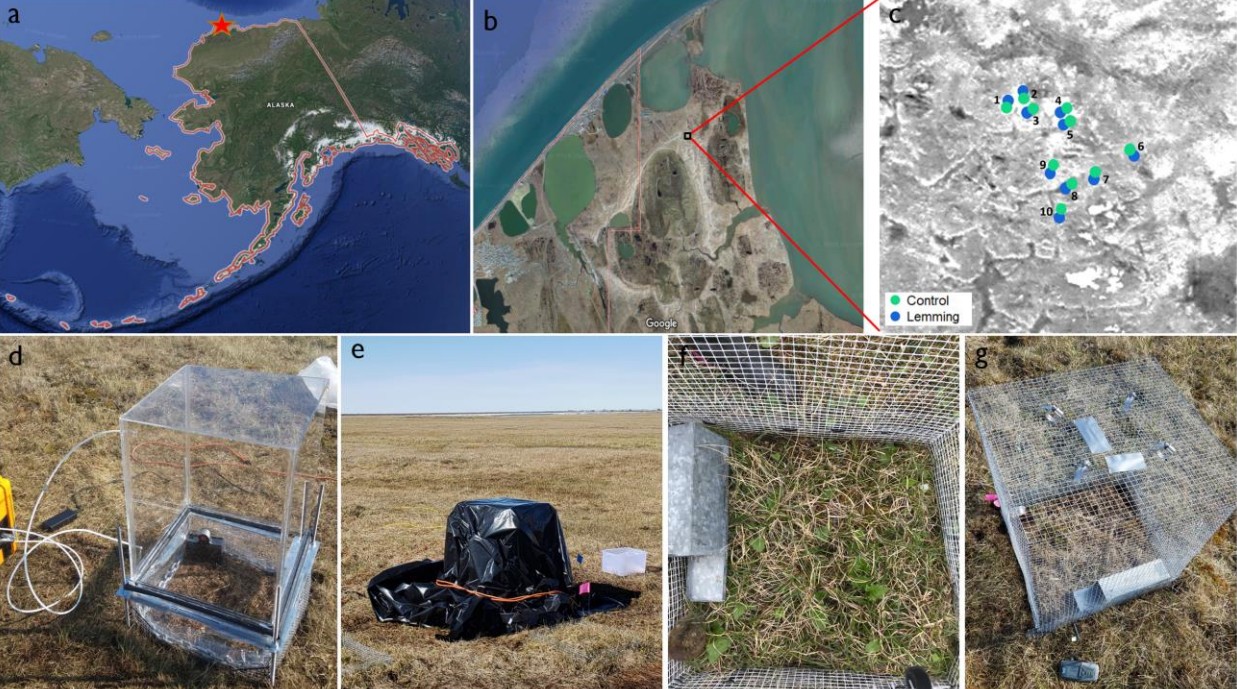


**Figure 1. (a)** The location of the study site, Utqiaġvik (Barrow), in Alaska (© Google Maps 2018, imagery from TerraMetrics) represented by a
red star, (**b**) location of the sampling site (© Google Maps 2018, imagery from TerraMetrics), (**c**) distribution of the sampling plots on an image
created using the coordinates of the plots in R (Worldview-3 panchromatic imagery taken 24 July 2016, Maxar Technologies), (**d**) chamber used
for the greenhouse gas flux measurements, (**e**) chamber covered by light-blocking material, and (**f**, **g**) enclosures installed at each of the plots
during the manipulation experiment.

## 2.2 Brown lemmings as a study species

Within the Arctic ecosystem of Alaska there are two species of lemmings: brown lemmings (*L. trimucronatus*) and northern collared lemmings (*Dicrostonyx groenlandicus*). Brown lemmings tend to be distributed among lower and middle Arctic tundra subzones (Stenseth, 1999). Although both brown and collared lemmings are found in Utqiaġvik, brown lemmings are more abundant in this region due to their preference for wetter habitats with relatively high-quality vegetation from lowlands (Batzli et al., 1983). Northern collared lemmings occupy drier habitats, and as a result are not as profuse and influential on vegetation in wet regions of the Alaskan Arctic such as Utqiaġvik (Batzli et al., 1983; Krebs et al., 2011; Morris et al., 2000).

### 2.2.1 Brown lemming consumption

Brown lemmings mostly consume graminoids in the summer and mosses in the winter (Batzli and Jung, 1980). Brown lemmings can consume much more than predicted by general trends of terrestrial vertebrates (EPPO, 1994), consuming up to eight times their body weight each day (Stenseth and Ims, 1993). Thus, their extreme capacity for consumption, combined with their cyclic elevated population densities in the region, can result in high vegetation removal. To wit, during winter lemmings destroy or uproot up to 90-100 % of surrounding aboveground biomass within their foraging range (Stenseth and Ims, 1993). Arctic vegetation consumed by lemmings is generally nutrient-poor (Batzli et al., 1980). Brown lemming digestive efficiency tends to be low, digesting only about 30 % of ingested food (Batzli et al., 1980). Due to consistent year-round activity and their small body size, lemmings also have a high metabolic rate. Low nutrient content, low digestive efficiency, and a high metabolic rate result in lemmings requiring a high rate of food intake for survival.

To reduce the risk of detection by predators (snowy owl, parasitic jaeger/arctic skua, arctic fox, and ermine), lemmings forage on small areas near their burrows and maximize their foraging in these areas until their primary food source is depleted, at which point they move to a new area of vegetation near a burrow or runway (Erlinge et al., 2011). This behavior shapes their foraging habits and leads to a higher concentration of grazing on vegetation close to burrows and runways (Erlinge et al., 2011). As a result, approximately 95-100 % of graminoid shoots are repeatedly clipped by lemmings occupying burrows and visiting runways in the immediate vicinity of the vegetation, and as the distance from the burrows and runways increases, clipping becomes patchier and the intensity of clipping on vegetation decreases (Batzli et al., 1980).

### 2.2.2 Brown lemming population

Populations of brown lemmings tend to reach peak densities every three to five years and then steeply decline (Stenseth, 1999). Interactions between lemming populations as fast-growing consumers and plant populations as slowly recovering resources represents a bitrophic system (Ims and Fuglei, 2005). In this system, vegetation could be heavily damaged by overgrazing during peak years of lemming abundance.

A report on the monitoring of lemming abundance and distribution (Ott and Currier, 2012) estimated brown lemming density near Utqiaġvik in 2012 to range from five to 65 lemmings per hectare. However, basic population density estimates may underestimate the impact lemmings have on some vegetation due to an increased concentration in grazing very close to burrows and runways (Erlinge et al., 2011). Ott and Currier (2012) also used baited Sherman traps to estimate abundance, a live-trapping technique that may lead to an underestimate of the actual population density for this species, as brown lemmings are not readily recaptured using baited Sherman traps; we found manual capture techniques to be much more effective than baited traps.

### 2.3 Sampling plan and experimental design

We performed this experiment over two summer growing seasons. During the first season (summer 2018), we used enclosures to
ensure even lemming herbivory pressure in each of our experimental plots. We manually captured the lemmings used in this first
season shortly after peak growing season (3-10 August), coinciding with accelerated lemming reproduction and peak population
density. We captured the lemmings in close proximity to our sampling sites while conducting visual encounter surveys, and secured
them in Sherman traps with cotton nestlets and vegetation (grasses and sedges). Our samples included both juvenile and adult life
stages. We released or avoided capturing any sick, very slow, or noticeably pregnant lemmings. After capture, we relocated
lemmings to the study site for inclusion in the experiment. Like voles (close relatives of lemmings), lemmings have distinct
preferences for habitats containing their preferred food items (Batzli and Henttonen, 1990), which we specifically selected for
when designing the location of the experimental plots in this study in order to represent realistic effects of lemming herbivory.
We established 10 plot sets for this experiment. Each of the 10 plot sets included a lemming plot paired to a control (no-
lemming) plot nearby (20 plots in 10 sets total) (Fig. 1c). Each plot contained different vegetation types (mosses, lichens,
graminoids, and wet sedges), and the control and lemming plot in each plot set was ensured to be as similar in composition as
possible in order to minimize biases due to spatial heterogeneity in vegetation and other landscape characteristics influencing
vegetation and carbon fluxes (a more in-depth analysis of these vegetation types was completed by Davidson et al. in 2016). We
placed control plots within 1 m of their paired lemming plot to keep environmental factors as similar as possible within plot sets;
we placed plot sets approximately 3 m away from each other. Plots were 50 x 50 cm in size; in each plot we dug a galvanized
hardware cloth with a ½ inch grid down through the thawed soil to the permafrost and up to 60 cm above the surface (Fig. 1f and
g). We selected the size of these plots to be consistent with a similar lemming exclosure experiment by Eskelinen and Virtanen
(2005) in Finland. This size was also similar, yet a bit larger than the experimental plots in the study by Lara et al. (2017) near
Utqiaġvik, Alaska which used 30 x 30 cm chamber bases within their exclosures. Control plots not only excluded lemmings for
the duration of the experiment, but also served as a control for the soil and vegetation disturbance resulting from digging galvanized
hardware cloth into the soil. Plots that included lemmings also included a top portion of hardware cloth that prevented lemmings
from escaping via climbing and prevented predators from removing the lemmings during the experiment. Inside each enclosure
with a lemming was a locked-open Sherman trap with cotton nestlets for protection from environmental elements.
Because rodents may experience physiological stress after being captured (Fauteux et al., 2018a), prior to the experiment we
kept the lemmings in small individual cages made of hardware cloth with a locked-open Sherman trap for shelter, cotton nestlets
for warmth, and vegetation for nutrition for at least an hour to help them acclimate. After this acclimation period, we placed the
lemmings in their individual plots for 16 hours. We based the duration of the experiment on field trials we carried out for several
weeks before the start of the experiment. These trials showed that 16 hours was enough time to observe an average impact on the
vegetation, visually similar to the effect lemmings have on areas near their burrows, but was not too long as to result in complete
vegetation consumption, unrepresentative of most areas where lemmings forage. Our field trials revealed that keeping lemmings
inside the enclosure for longer than 16 hours (which varied with lemming size) led to a complete vegetation removal, an extreme
scenario only observed in the very close proximity of the burrows, and not representative of most of their foraging areas. We
released the lemmings at the end of all these experiments in proximity to the locations where they were captured.
The subsequent season (summer 2019), we re-visited the sample plots to measure the impact of lemmings one year after their
grazing (24 June-7 August). During this season, we did not capture any lemmings, nor did we perform any additional manipulation.
To be able to assess longer-term impacts of the manipulations carried out the previous summer, we collected measurements
throughout the following summer (Table 1) to represent pre-, early, and peak growing season (hereafter defined as "rounds").
Sampling was carried out to monitor the timing of regrowth of photosynthetic tissue and recovery of the plants at different times

of the season in 2019: 24-29 June (round one: pre-growing season), 9-19 July (round two: early growing season), and 29 July-7 August (round three: peak growing season).

| | | Data Collected | Frequency of Measurement |
|---|---|---|---|
| **Summer 2018** | | $CO_2$ fluxes (NEE) and $CH_4$ fluxes, NDVI, air temperature, soil temperature, soil moisture, thaw depth, motion camera footage | Before (pre-herbivory) and after (post-herbivory) lemming treatment (N=10 before and N=10 after for each treatment, for a total of N=40 per NEE, $CH_4$ fluxes, NDVI; N=40 for air temperature; N=40 for soil temperature; N=100 for soil moisture; N=20 for thaw depth) |
| **Summer 2019** | | $CO_2$ fluxes (NEE, ER, GPP) and $CH_4$ fluxes, NDVI, air temperature, soil temperature, soil moisture, thaw depth | Different times of the season (pre-, early, peak growing season) (N=10 for each round and treatment, for a total of N=60 per NEE, ER, GPP, $CH_4$ fluxes, NDVI; N=60 for air temperature; N=300 for soil temperature; N=240 for soil moisture; N=240 for thaw depth) |

**Table 1.** Types of data collected and when they were measured during summer 2018 and summer 2019. All data were collected while lemmings were not present inside the experimental plots, except for the motion camera footage. NEE is defined as net ecosystem exchange, ER as ecosystem respiration, GPP as gross primary production, and NDVI as the normalized difference vegetation index.

There could have been other sources of herbivory (such as caribou), but these sources are not as frequent in these northernmost areas of the Arctic Coastal Plain. Additional sources of disturbance to vegetation could have originated from a drastic change in environmental conditions, such as extreme temperatures, extremely dry conditions, etc.; however, these would not have selectively removed the vascular plants while not affecting the moss layer, which is what we observed in this experiment.

**2.4 Greenhouse gas fluxes and environmental variables measurements**

We used a Los Gatos Research (LGR) Ultraportable Greenhouse Gas Analyzer (UGGA Model 915-0011) to measure $CO_2$ and $CH_4$ concentrations (currently, global mean atmospheric concentrations for $CO_2$ and $CH_4$ are approximately 417 ppm and 1909 ppb, respectively; NOAA GML, 2022) and air temperature over time in all plots during both summer seasons (2018 and 2019). We measured $CO_2$ and $CH_4$ concentrations one day after lemming removal from the plots in summer 2018 (exact time varied based on weather conditions and when plots were measured in temporal relation to other plots) and during the different rounds of the growing season in summer 2019. To collect measurements, we built a clear plexiglass acrylic chamber (Davidson et al., 2016; McEwing et al., 2015) to enclose the plots once the aboveground portion of the caging had been detached and the lemming had been removed (Fig. 1d). This chamber was placed on a metal frame positioned in the ground outside of the plots and had clear polyvinyl material weighed down by heavy metal chains to produce a seal inside the chamber. These measurements were performed in a closed loop, where tubes connected the chamber to the gas analyzer and then air was circulated back to the chamber. We positioned a small fan inside the chamber to ensure appropriate air mixing. We collected greenhouse gas concentrations in the absence of lemmings.

We used the rate of concentration change to calculate carbon fluxes using the chamber volume and area covered by vegetation (i.e., responsible for the carbon emission or uptake) as a function of time, as described in McEwing et al. (2015). The $CO_2$ concentration change allowed us to calculate the net balance between the carbon uptake from photosynthesis and the carbon loss from respiration, also defined as the net ecosystem exchange (NEE), before and after the first summer's manipulations (2018), as previously described, and to track the seasonal development of NEE during the second summer (2019). In the second summer, we used a light-blocking material to cover the chamber (Fig. 1e) for determining $CO_2$ ecosystem respiration (ER) and gross primary production (GPP) from NEE, calculated following Eq. (1):

$$GPP = NEE - ER \,, \tag{1}$$

and using the sign convention suggested by Chapin et al. (2006). Since plant growth and photosynthetic uptake is restricted to the
summer months in these Arctic ecosystems, we used GPP to indicate "the total amount of $CO_2$ 'fixed' by land plants per unit time
through the photosynthetic reduction of $CO_2$ into organic compounds" (Gough, 2011) during the time of measurements, rather than
as an annual measurement.
We also measured a variety of environmental variables before and after each portion of the experiment (summer 2018) and
during each round (summer 2019). These environmental variables included air temperature recorded by the LGR gas analyzer, soil
temperature measured with a Thomas Scientific Traceable Kangaroo thermometer, soil water content recorded by a FieldScout
Soil Moisture Meter, and thaw depth using a metal probe marked every 5 cm. We examined these variables as controls that may
explain shifts in $CO_2$ and $CH_4$ fluxes within the study area and to monitor if plots in each of the sets experienced similar abiotic
conditions. This assured potential differences in carbon fluxes were due to our manipulation, and not different environmental
conditions of various plots.

## 2.5 Camera footage and hyperspectral measurements

We quantified the impact of lemming herbivory and burrowing on vegetation using a Spectra Vista Corporation (SVC)
Spectroradiometer HR-512i, which measured spectral reflectance and recorded a picture of the vegetation being scanned. The
spectrometer yielded hyperspectral measurements for vegetation in the 338.9-1075.1 nm spectral range with 512 bands and a
bandwidth of $\leq 1.5$ nm. We used the internal global positioning system (GPS) of the spectroradiometer to record geographic
coordinates (latitude and longitude) for all plots to an accuracy of 2.5 m. We collected hyperspectral measurements in the absence
of lemmings.
We measured total reflected spectral exitance from a blank white reference panel right before sampling each plot set
(approximately every 20-30 scans, or 10-15 minutes) to estimate spectral irradiance based on reflectance calibration information
provided for the reference panel. We recorded spectral surface reflectance before and after each experimental treatment (summer
2018) and at different times during the season in the following summer (2019) and used it to calculate narrow-band normalized
difference vegetation index (NDVI) to compare the photosynthetic capacity of vegetation in the plots. NDVI is calculated as the
normalized difference between reflectance in the near infrared wavelengths (800.5 nm) and red wavelengths (680.2 nm). Lower
values of NDVI indicate no living vegetation and higher values indicate more green biomass.
We recorded a time-lapse of various parts of the experiment using a Brinno MAC200DN Outdoor Camera to collect motion-
sensor video footage of lemming activity. The camera also allowed for re-visitation and surface cover characterizations of the plots
to classify and quantify vegetation types within each plot and assess how grazing had affected vegetation. We did not systematically
record all trials on video, but instead used this technology as a qualitative tool to visually document the activity of the lemmings.

## 2.6 Statistical analyses

We used the statistical program R, version 3.5.1 (R Core Team, 2019), for our statistical analyses. We tested for normality using
a Shapiro-Wilk normality test. The 2018 data were normally distributed (NEE $P = 0.489$, NDVI $P = 0.816$), except the $CH_4$ data
($P < 0.001$), which were right-skewed, so we log-transformed the $CH_4$ data to help normalize them ($P = 0.284$). After this
transformation, we used linear mixed-effects models (with the package "nlme"; Pinheiro et al., 2018) to test for the significance
between the different treatments. For the 2019 data, we used both linear mixed-effects models and non-parametric Kruskal–Wallis
tests because we could not make all the data normal using the same transformation method (log transformation, square root
transformation) for every round during the season. We also tested for equal variance using an F-test and found there was no
significant difference between the variances (treatments) in 2018 (NEE $P = 0.172$, $CH_4$ flux $P = 0.810$, NDVI $P = 0.100$) and 2019
(NEE $P = 0.441$, ER $P = 0.650$, GPP $P = 0.852$, $CH_4$ flux $P = 0.346$, NDVI $P = 0.951$).

248       We tested multiple variations of the linear mixed-effects models using the methods for model selection in ecology described

in Zuur et al. (2009). We then plotted the models to examine the residuals of the data and found them to not appear heteroscedastic.
For the 2018 models, we used treatment (control, lemming plots), time (before, after experiment), and their interaction as fixed
factors in the models; for the 2019 models, we used treatment (control, lemming plots), round (pre-, early, and peak growing
season), and their interaction as fixed factors in the models. In all analyses we used the plot identification (1C, 1E, 2C, 2E, etc.)
nested within the plot set (1-10) as random factors. Mixed models allow us to account for temporal and spatial pseudoreplication
and to test the significance of the interactions among factors. When fixed factors were significant, we conducted a pairwise analysis
via a Tukey post-hoc test (with the package "emmeans"; Lenth et al., 2019) to investigate the interacting effects in the model.

256       To identify the effect of the manipulation on carbon fluxes and NDVI, we applied the linear mixed-effects models and tested

for differences in each environmental variable before and after lemming exposure in summer 2018. We then used the statistical
analyses to help us explore if the post-lemming experimental plots showed a significant change in carbon fluxes and NDVI when
compared to pre-lemming experimental plots (2018), and if the carbon fluxes and NDVI varied between treatments the following
growing season (2019). Our analysis of the NDVI from spectral indexes provided us with information on changes in plant biomass
before and after each manipulation in summer 2018 and vegetation regrowth in summer 2019.
**3 Results**
**3.1 Environmental variables**
Environmental controls on $CO_2$ and $CH_4$ fluxes such as air temperature, soil temperature, thaw depth, and soil moisture were
similar between the control and experimental plots in 2018 (Fig. A1a-h) and 2019 (Fig. A1i-p). During summer 2018, air
temperature ($P = 0.542$), soil temperature ($P = 0.960$), thaw depth ($P = 0.683$), and soil moisture ($P = 0.619$) were not significantly
different between control plots and lemming plots; during summer 2019, air temperature ($P = 0.887$), soil temperature ($P = 0.060$),
thaw depth ($P = 0.512$), and soil moisture ($P = 0.387$) were not significantly different between the control and lemming treatments.
**3.2 Carbon fluxes**
The presence of lemmings significantly impacted $CO_2$ fluxes (i.e., NEE) during summer 2018 when the lemming enclosure
treatment was implemented. Before the treatment, calculated NEE (Fig. 2a) and $CH_4$ fluxes (Fig. 3a) for the control and lemming
plots were similar. After the lemming experiment (and removal of the lemmings from the experimental plots), the net $CO_2$ uptake
decreased significantly ($P < 0.001$, Fig. 2a). In this context, net $CO_2$ uptake by vegetation, or carbon dioxide sequestration, was
the removal of $CO_2$ from the atmosphere and its storage in the above- and belowground biomass through photosynthesis after
accounting for the carbon loss through respiration. Therefore, when lemmings consumed the photosynthetic tissues of the
vegetation (aboveground biomass), the vegetation was no longer able to uptake $CO_2$ from the atmosphere, and NEE (the net
ecosystem exchange equivalent to the net $CO_2$ fluxes) approached either zero or became less negative (a negative sign implies
more carbon removal from the atmosphere). By the end of summer 2018, the effect of brown lemmings' herbivory changed the
mean NEE for lemming plots from $-0.074 \pm 0.012$ gC-$CO_2$m$^{-2}$h$^{-1}$ (i.e., net $CO_2$ sequestration) to $0.003 \pm 0.012$ gC-$CO_2$m$^{-2}$h$^{-1}$ (i.e.,
net $CO_2$ fluxes were around zero). Contrary to what we expected, $CH_4$ flux values did not significantly differ between control plots
and plots subjected to lemmings' herbivory ($P = 0.989$, Fig. 3a).

In summer 2019, we measured NEE and $CH_4$ fluxes again, and additionally calculated ER and GPP. During this second summer of measurements, results of the linear mixed-effects models for NEE, ER, GPP, and $CH_4$ fluxes were all not significantly different between control and lemming plots (NEE $P = 0.834$, Fig. 2b; ER $P = 0.742$, Fig. 4a; GPP $P = 0.716$, Fig. 4b; $CH_4$ flux $P = 0.869$, Fig. 3b). These results were consistent with those of the Kruskal–Wallis test, which found there was no significant difference between the treatments in 2019, either by testing the data set all together (NEE $P = 0.769$, ER $P = 0.221$, GPP $P = 0.513$, $CH_4$ flux $P = 0.824$) or separating it for different times of the season (rounds) and testing each time separately (pre-growing season: NEE $P = 0.245$, ER $P = 0.672$, GPP $P = 0.296$, $CH_4$ flux $P = 0.728$; early growing season: NEE $P = 0.853$, ER $P = 0.600$, GPP $P = 0.558$, $CH_4$ flux $P = 0.638$; peak growing season: NEE $P = 0.293$, ER $P = 0.366$, GPP $P = 0.212$, $CH_4$ flux $P = 0.970$).

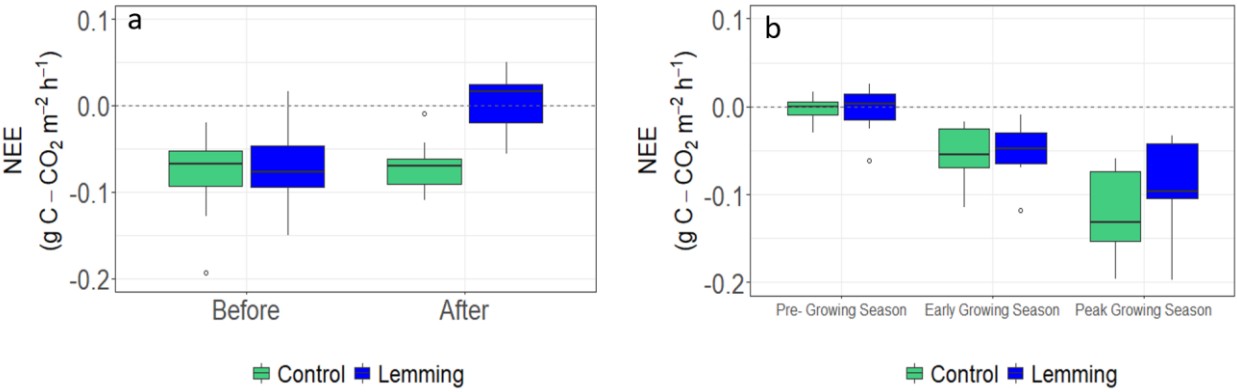

**Figure 2.** Box and whisker plots of 2018 and 2019 NEE for the control and lemming plots. Negative flux values indicate carbon sequestration/uptake from the atmosphere by vegetation though photosynthesis and positive flux values indicate carbon emission/loss into the atmosphere. (**a**) Median NEE for plots before and after the experiment in summer 2018 ($T = 4.62$, $P < 0.001$), and (**b**) median NEE for plots during the three rounds of measurements in summer 2019 ($T = 0.21$, $P = 0.834$).

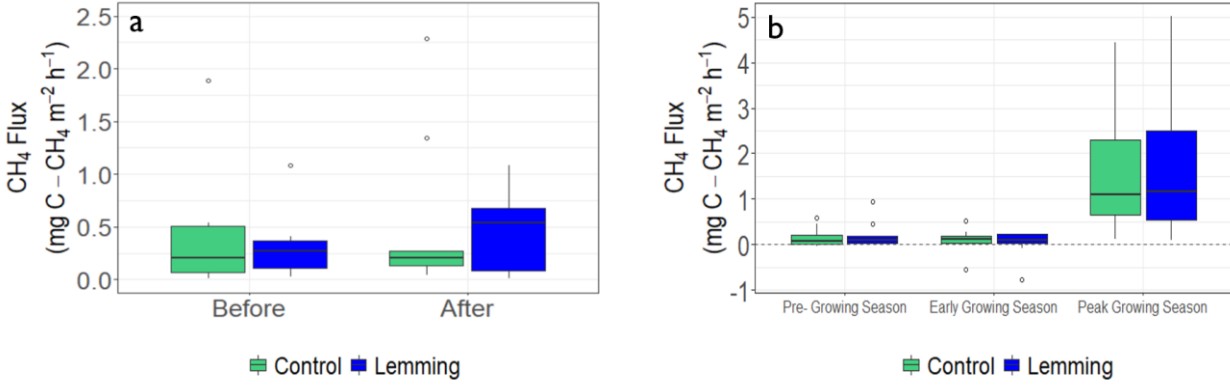

**Figure 3.** Box and whisker plots of 2018 and 2019 $CH_4$ fluxes for control and lemming plots. Negative flux values indicate uptake from the atmosphere and positive flux values indicate emission to the atmosphere. (**a**) Median $CH_4$ flux for plots before and after the experiment in summer 2018 ($T = 0.01$, $P = 0.989$), and (**b**) median $CH_4$ flux for plots during the three rounds of measurements in summer 2019 ($T = -0.17$, $P = 0.869$).

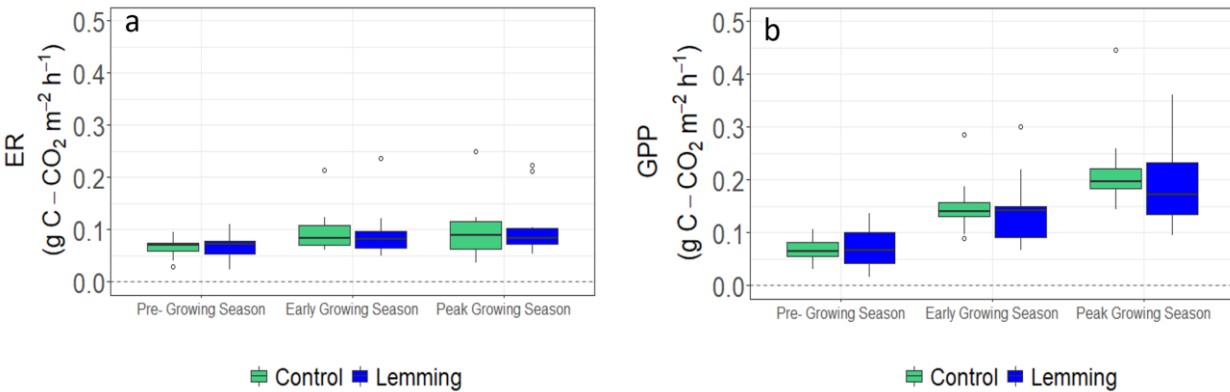

**Figure 4.** Box and whisker plots of ER and GPP for control and lemming plots during the three rounds of data collection in summer 2019. Positive flux values indicate a positive respiration (carbon loss into the atmosphere) and a positive carbon uptake by vegetation through photosynthesis. (**a**) Median ER ($T$ = -0.34, $P$ = 0.742), and (**b**) median GPP ($T$ = -0.37, $P$ = 0.716). The signs of ER and GPP are always positive, but if ER is more than GPP, then the ecosystem is a carbon source into the atmosphere (with a positive sign of NEE).

### 3.3 Hyperspectral surface reflectance and NDVI

Spectral reflectance derived from spectroradiometric radiances generally increased across visible and near infrared wavelengths after lemmings' vegetation removal (Supplementary Fig. S1). Before placing lemmings in enclosures, control and lemming plots exhibited similar surface reflectance values, while reflectance curves showed more substantial separation after lemming removal (Supplementary Fig. S1). Analyzing the surface reflectance of the same control and lemming plots re-visited in summer 2019 revealed that the reflectance values for these different treatments were alike in each plot set, similar to what was observed before the beginning of the manipulation experiment (Supplementary Fig. S2b).

To better quantify the changes in reflectance, we calculated the NDVI in all the control and treatment plots in both summer 2018 and 2019. Following lemming removal in the first summer, lemming plots had significantly lower NDVI than the control plots *(P* = 0.015, Fig. 5a), consistent with the decrease in green biomass observed in the photographs collected before and after placing the lemmings in the treatments' enclosure (Supplementary Fig. S1), and with the decreases in net $CO_2$ uptake (see NEE close to zero after lemming vegetation consumption; Fig. 2a). The effect of brown lemmings' herbivory changed the mean NDVI for lemming plots from 0.551 ± 0.021 to 0.465 ± 0.021. During the second summer, median NDVI values of all plots were similar (Fig. 5b). Results of the linear mixed-effects model reveals that during this time, there was no significant difference in NDVI when comparing control plots to lemming plots ($P$ = 0.692), which is consistent with the results of the Kruskal–Wallis test that found no significant difference between the treatments in 2019 either by testing the data set all together ($P$ = 0.694) or separating it for different times of the season (rounds) and testing each time separately (pre-growing season: $P$ = 0.260, early growing season: $P$ = 0.418, peak growing season: $P$ = 0.283). There was a significant difference in NDVI across the rounds ($P < 0.001$), which coincides with the increased green biomass observed in collected photographs from pre- to early to peak growing season (Supplementary Fig. S2a).

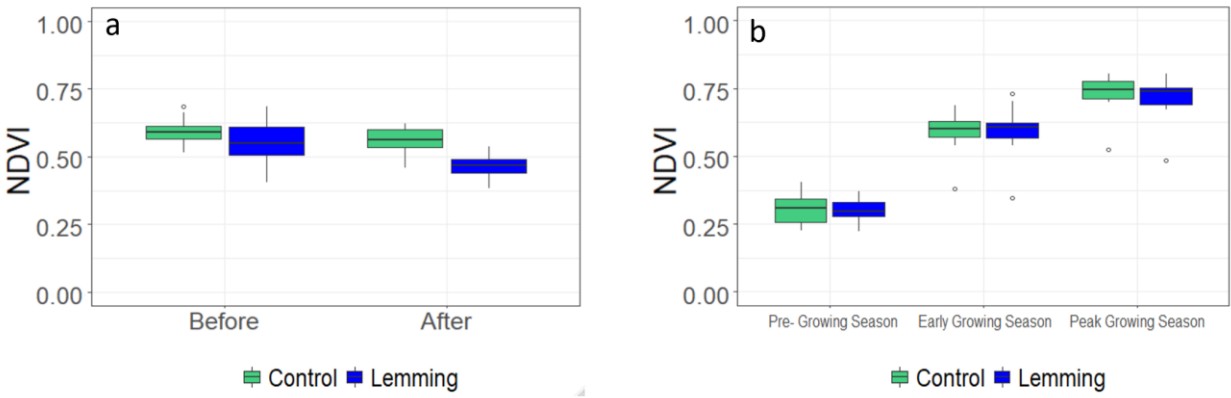

324

**Figure 5.** Box and whisker plots of 2018 and 2019 NDVI values for control and lemming plots. (**a**) Median NDVI for plots before and after the experiment in summer 2018 ($T = -3.69$, $P = 0.015$), and (**b**) median NDVI for plots during the three rounds of measurements in summer 2019 ($T = 0.41$, $P = 0.692$). Higher vascular plant green biomass in the pre-lemming treatment plots presented NDVI values in the 0.6 to 0.7 range, whereas post-lemming treatment plots in 2018 exhibit NDVI values around 0.5.

## 4 Discussion

We found, within a short-term enclosure experiment, that brown lemmings' herbivory significantly decreased net $CO_2$ uptake immediately after consumption of vegetation, while surprisingly not affecting $CH_4$ fluxes. Consumption of photosynthetically active plant tissue by lemmings impacted the ability of the vegetation to sequester $CO_2$, nullifying $CO_2$ uptake by tundra vegetation. The lack of significant difference in the evaluated environmental variables (air temperature, soil temperature, soil moisture, thaw depth) between the control and lemming treatment plots suggests that these factors did not play a significant role in the difference in net $CO_2$ fluxes before and after the treatments during the first summer. Therefore, we assume that the vegetation removal was the main reason for the decrease in the ability of the ecosystem to sequester carbon. Unfortunately, the design of this experiment, mostly focusing on the aboveground measurements (except for the soil temperature, soil moisture, and thaw depth), did not allow for identifying the contribution of belowground increased decomposition from the aboveground vegetation removal.

Notably, lemmings' herbivory did not affect $CH_4$ fluxes, even though sedges have an important role in facilitating $CH_4$ transport from deeper soil layers (and ultimately emissions into the atmosphere) in tundra ecosystems (Lai, 2009; McEwing et al., 2015), and also provide substrate for methanogenesis, which should increase $CH_4$ production and emission (Bridgham et al., 2013). The lack of a significant effect on $CH_4$ fluxes may have been due to the location of vegetation removal on consumed plants. Kelker and Chanton (1997) showed the location of the clipping of vegetation affects the $CH_4$ emissions; belowground clipping at the root-shoot or porewater-root boundary is more likely to impact $CH_4$ emission, but aboveground clipping is less likely to affect $CH_4$ emission. This differential effect is likely related to the location of $CH_4$ escape though vegetation, which is just at the root-shoot or porewater-root boundary (Kelker and Chanton, 1997). Moreover, vegetation can have an impact on stimulating $CH_4$ through labile carbon exuded by the roots (McEwing et al., 2015; Ström et al., 2003; Zona et al., 2009). Methanogenesis is fueled by labile carbon, aiding in $CH_4$ production in the Arctic (Tan et al., 2015). Labile carbon released by root exudation depends on photosynthetic activity of vegetation and ultimately stimulates $CH_4$ emission (Bhullar et al., 2014; Ström et al., 2003). The lack of response of $CH_4$ emissions to vegetation removal could be explained by the large soil carbon stored in these permafrost soils (Hugelius et al., 2014). A decrease in labile carbon exudation due to vegetation removal from herbivory may have not been limiting $CH_4$ emissions, consistent with a lack of response in $CH_4$ emissions with a labile carbon addition in these sites (von Fischer et al., 2007; Zona et al., 2009).

Moreover, when measured shortly after the lemming treatment, the $CH_4$ emission in the plots may have been inhibited by lemming urine. Ammonium from urine has been linked to an increase in $CH_4$ production (Lin et al., 2009); however, it has been found that $CH_4$ fluxes can initially result in a mean negative flux shortly after the addition of urine to the system (Boon et al., 2014). The timing in which we measured the greenhouse gases after the lemming treatments falls within the initial window of time found by Boon et al. (2014) to have this effect; thus, urine produced by the lemmings in the plots may have nullified the positive $CH_4$ emissions via the aerenchyma. Without further investigation into the soil chemistry, it is difficult to determine which mechanisms of herbivore-plant interactions resulted in the lack of significance in $CH_4$ emission.

As expected, the biomass of vegetation decreased during summer 2018 due to the impact of lemming consumption (Supplementary Fig. S1). The control and experimental plots before the lemming treatment had relatively high and similar mean NDVI values (Supplementary Fig. S1), suggesting their biomass had similar values (Goswami et al., 2015). Vegetation removal by brown lemmings significantly lowered the mean NDVI of the plots subjected to lemming herbivory. By summer 2019, the mean NDVI value of these same lemming plots indicated that the vegetation was fully recovered from the lemmings' impact the previous summer. Measurements collected the summer following our herbivory experiment (2019), revealed that the vegetation recovery after brown lemming disturbance was rapid and quickly regrew to a condition comparable to that found in 2018, prior to lemming consumption. Since lemmings mostly consume vascular plants, such as graminoids and sedges, in the summer and avoid non-vascular and slower growth vegetation, such as mosses and lichens (Batzli et al., 1980), the preferential consumption of annual grasses and sedges likely led to the rapid recovery of the photosynthetic capacity of vegetation we observed in just one year. From analysis of the motion-sensor video footage, we observed lemming foraging within the plots was representative of these vegetation preferences. This is consistent with the vegetation being mostly dominated by grasses and sedges in the sites of this research (Davidson et al., 2016).

While our experiment showed a potentially substantial impact of lemming herbivory on the $CO_2$ fluxes from these tundra ecosystems, we did not address the impact of varying degrees of intensity of herbivory and population cycling of brown lemmings on carbon fluxes and photosynthetic capacity of different vegetation communities. Roy et al. (2020) found that herbivore presence can alter communities of vegetation differently, as herbivores play a role in regulating a variety of plant species. These herbivores can lead to significant changes in the abundance of vegetation types, allowing for the potential of the tundra during the peak growing season to switch between a carbon source to sink in the absence of herbivory (Min et al., 2021). Since brown lemmings rely on a high rate of food intake to sustain growth and reproduction (Batzli et al., 1980) and experience population cycles with distinct seasonal and multiannual density fluctuations (Reid et al., 1995; Stenseth, 1999), rapid consumption of plant matter by lemmings as sustenance during population peaks may significantly contribute to shifts in plant communities and, thus, carbon cycle changes.

Since lemming population densities vary in response to multiple environmental factors (Fauteux et al., 2015; Soininen et al., 2017), predicting a 'normal' level of herbivory for this species is very challenging. Reports on estimated brown lemming density have found their local density to range from five to 65 lemmings per hectare (Ott and Currier, 2012; Alaskan Arctic) and about zero to nine lemmings per hectare (Fauteux et al., 2015; Canadian Arctic), which is variable and may be an underestimate due to the use of live-trapping, as mentioned previously. Moreover, in addition to space, it is important to consider time: we only kept lemmings inside the plots for 16 hours and there was no effect of lemming herbivory for the remainder of the experiment. The most relevant comparison we could find to define the degree of herbivory observed was the effect on vegetation near lemming burrows and runways in a similar ecosystem (e.g., Siberian tundra; Erlinge et al., 2011). Given the sparsity of available literature and data from these understudied Arctic ecosystems, it is difficult to categorize our lemming treatment as having some sort of 'normal' or 'heavy' impact on vegetation, which would be required to explore legacy effects of lemming herbivory.

Lemming populations may also vary in response to regulation by predators (Fauteux et al., 2018b), and predation risk may
change lemming physiological response and foraging behavior (Hawlena and Schmitz, 2010). In many terrestrial systems, indirect
effects of predator presence on herbivores have been shown to have dramatic effects on vegetation consumption (Apfelbach et al.,
2005; Borowski, 1998), with resulting behavioral changes rippling through the ecosystem (Ripple and Beschta, 2003). Given the
substantial impact of lemming herbivory on the tundra carbon balance, indirect cues indicating predator presence may alter
lemming behavior and thus vegetation. If predator cues elicit a fear response in the lemmings, therefore decreasing the time spent
consuming vegetation, this change in behavior may decrease the severity of lemmings' impact on vegetation and carbon cycling,
specifically their negative affect on $CO_2$ sequestration. The influence of predator-prey interactions on herbivory, and how they
further impact vegetation and carbon fluxes in the Arctic tundra should be quantified by future studies to better understand
multifaceted interactions in the Arctic (see supplementary materials).

## 5 Conclusions

We show that there is an immediate effect of lemmings on plant biomass and net $CO_2$ uptake by Arctic vegetation, but not on $CH_4$
fluxes in areas where lemmings forage. However, impacts on vegetation are temporary, and plant biomass and net $CO_2$ uptake can
recover to previous conditions by the end of the subsequent growing season. To further our understanding of the complex
interactions in the Arctic, it is vital to also explore the longer-term feedbacks that may exist between climate change, herbivory,
and predator-prey interactions. The effects of warming on snow cover and plant growth, as crucial environmental resources to
lemmings, could lead to drastic population changes for lemmings, and the longer-term effect of lemmings' herbivory on vegetation
might not be captured by a short-term manipulation. It is also critical to link the long-term lemming population fluctuations to
potential shifts in vegetation and climate change. Additionally, climate change is likely to also alter the abundance, behavior, or
even occurrence of predators of lemmings, which may in turn impact lemming abundance and foraging behaviors. For these
purposes, longer-term and broader scale ecological data would be particularly valuable to build on the short-term effects
highlighted in this study.
Overall, our study suggests that brown lemmings have the ability to significantly alter vegetation by consuming photosynthetic
tissue, which hinders carbon sequestration by the vegetation and shifts $CO_2$ fluxes in the areas surrounding their burrows and
runways. We report that this effect is short-lived due to the preferential consumption by lemmings of plant species that quickly
regrow and recover by the next growing season. However, the duration of the impacts of lemming herbivory may change in
different vegetation communities, as various plant species might be affected differently. Thus, it is relevant to examine the effects
of lemmings on a wide range of ecosystems to make regional estimates of their short-term influence on net $CO_2$ fluxes and NDVI.
Future research should also more carefully quantify the interactions between lemmings, their predators, and carbon cycling in the
Arctic tundra ecosystem, which might explain some of the substantial interannual variability in the tundra $CO_2$ fluxes not explained
by environmental variables alone.
*Code availability.* R codes generated for data analysis during this study will be archived to the Arctic Data Center by the corresponding author
upon the journal's request.
*Data availability.* Data on carbon fluxes, hyperspectral surface reflectance, and environmental variables analyzed during this study will be
archived to the Arctic Data Center by the corresponding author upon the journal's request. All relevant data are included as figures in the paper,
and raw data may be made available upon request.

*Author contribution.* Study conception and design were carried out by Jessica Plein, Rulon Clark, Walter Oechel, and Donatella Zona. Material preparation, data collection, and data processing were completed by Jessica Plein. Data scripts and codes were written by Jessica Plein, Kyle Arndt, and Donatella Zona. Data analysis was performed by Jessica Plein and Donatella Zona. The drafts of the manuscript were written by Jessica Plein and all authors commented on previous versions of the manuscript. All authors read and approved the final manuscript.

*Competing interests.* The authors declare that they have no conflict of interest.

*Acknowledgements.* We thank the Global Change Research Group for equipment use, field support, and suggestions on the project design. We also thank Nicholas Barber for help with statistical analyses, George Aguiar at Archipelago Farms for reindeer urine collection (see supplementary materials), and Lupita Barajas, Marco Montemayor, Thao Tran, and Brian Graybill for their efforts with field set-up/take-down and measurements. The authors would like to thank the Polar Geospatial Center for the geospatial support. We express gratitude to NOAA for providing access to their observatory site, and especially Bryan Thomas for his expertise and guidance at the site. We thank the Ukpeaġvik Iñupiat Corporation (UIC) for the use of their laboratory, radios, and storage. The authors are also thankful to the Iñupiat people for sharing their culture and home with us. Welfare of animals indicated in Alaska Department of Fish and Game permits 18-167 and 19-131 and Institutional Animal Care and Use Committee (IACUC) Animal Protocol Form #16-08-014C.

*Financial support.* This study was supported by the Office of Polar Programs of the National Science Foundation (NSF) (award numbers 1204263, 1702797, and 1932900) with additional logistical support funded by the NSF Office of Polar Programs, by the NASA ABoVE (NNX15AT74A; NNX16AF94A) Program, by the European Union's Horizon 2020 research and innovation program under grant agreement No. 727890, by the Natural Environment Research Council (NERC) UAMS Grant (NE/P002552/1), and by the NOAA Cooperative Science Center for Earth System Sciences and Remote Sensing Technologies (NOAA- CESSRST) under the Cooperative Agreement Grant # NA16SEC4810008.

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

# B51C-0609, 2007.
Whiting, G. J. and Chanton, J. P.: Primary production control of methane emission from wetlands, Nature, 364, 794-795,
https://doi.org/10.1038/364794a0, 1993.
Zona, D., Oechel, W. C., Kochendorfer, J., Paw U, K. T., Salyuk, A. N., Olivas, P. C., Oberbauer, S. F., and Lipson, D. A.: Methane
fluxes during the initiation of a large-scale water table manipulation experiment in the Alaskan Arctic tundra, Global
Biogeochem. Cy., 23, GB2013, https://doi.org/10.1029/2009GB003487, 2009.
Zona, D., Oechel, W. C., Peterson, K. M., Clements, R. J., PAW U, K. T., and Ustin, S. L.: Characterization of the carbon fluxes
of a vegetated drained lake basin chronosequence on the Alaskan Arctic Coastal Plain, Glob. Change Biol., 16, 1870-1882,
https://doi.org/10.1111/j.1365-2486.2009.02107.x, 2010.
Zuur, A. F., Ieno, E. N., Walker, N. J., Saveliev, A. A., and Smith, G. M.: Mixed effects models and extensions in ecology with R,
Springer, New York, NY, 2009.
**Appendix A**

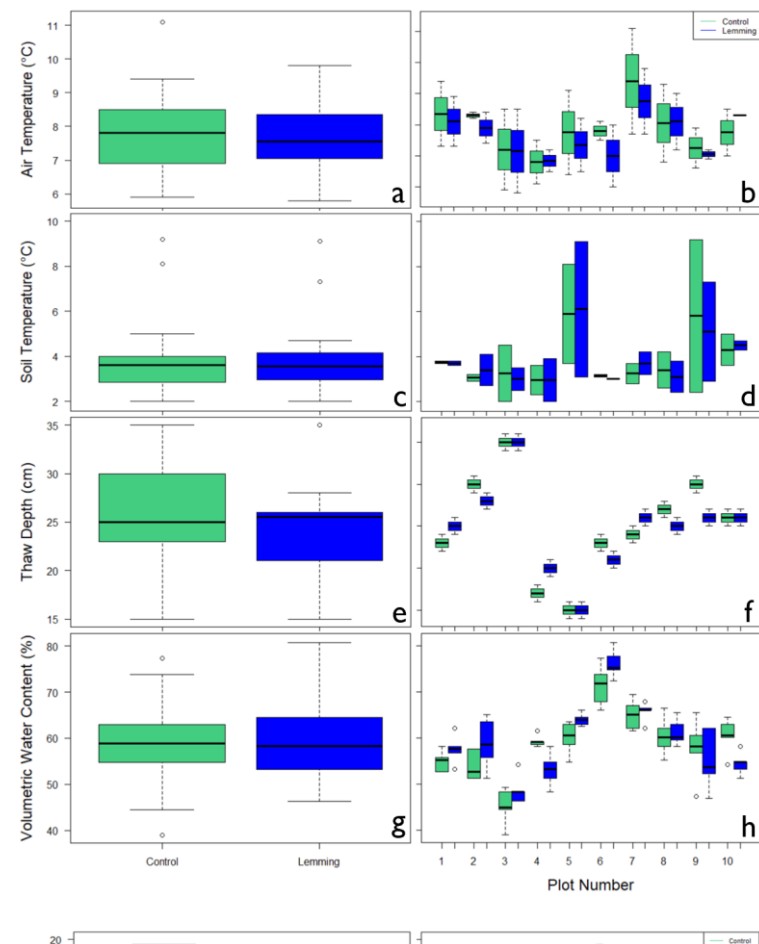


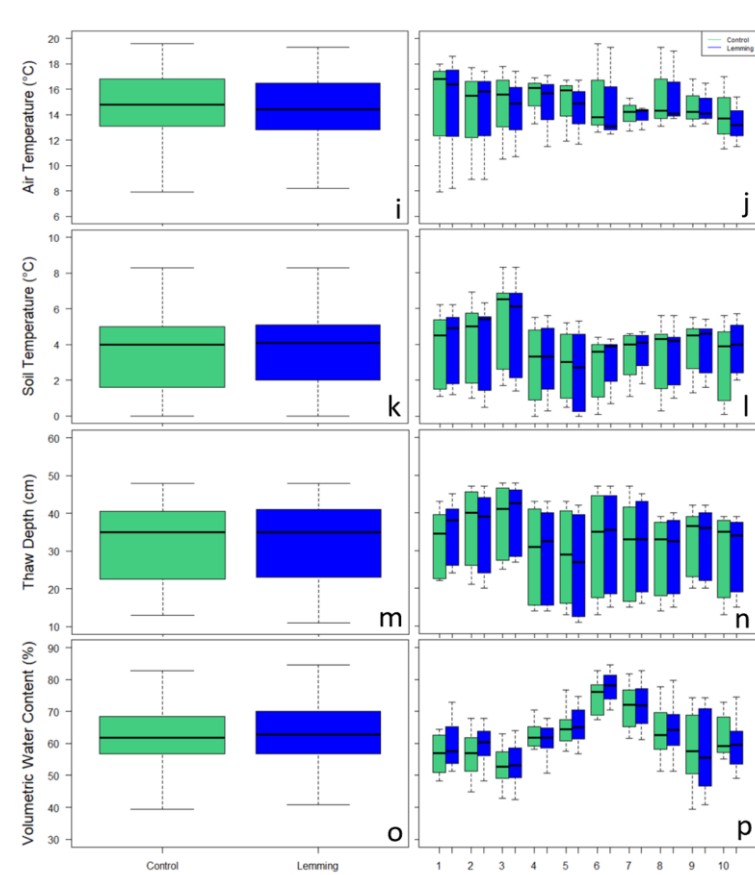


**Figure A1.** Box and whisker plots of environmental variables across treatment plots during (**a-h**) summer 2018 and (**i-p**) summer 2019.
Environmental variables include (**a**, **i**) air temperature for the entire dataset, (**b**, **j**) air temperature by plot, (**c**, **k**) soil temperature for the entire
dataset, (**d**, **l**) soil temperature by plot, (**e**, **m**) thaw depth for the entire dataset, (**f**, **n**) thaw depth by plot, (**g**, **o**) soil moisture for the entire dataset,
and (**h**, **p**) soil moisture by plot.