# Peer review of "Response of vegetation and carbon fluxes to brown lemming herbivory in Northern Alaska"

_Biogeosciences, 2021_

## Referee Comment (RC2)

[revised manuscript text omitted]

*Margin annotations:*

would (line 37) our (line 37)

awkward phrasing 41/42; also redundant with 40

compared to other ecosystems where? other non-tundra systems in the Arctic, or elsewhere globally where lemmings exist?

but see

What is the crucial gap that you're taking on here? I think there needs to be a sentence between the last two in this paragraph that spells it out explicitly for those of your readers who aren't Arctic experts (like myself!).

exclosures? are you manipulating population density? etc.! that'd be why?

short pulses of? I'm not totally clear on how enclosures show you herbivory if lemmings are already present, and I think it'd be good to emphasize this.

sequestration in soil or in veg?

soil ^

emission to decrease with decrease in biomass of vascular plants, given their role in the $CH_4$ transport from deeper anoxic soil layers into the atmosphere. Our broader goals are to increase our understanding of how the foraging behaviors of these herbivores impacts carbon dioxide ($CO_2$) and methane ($CH_4$) fluxes and the photosynthetic capacity of plants in the Alaskan Arctic

*introduce the shorthand for these gases the first time you use them in the intro*

environment.

*I think I'd more clearly delineate your hypotheses at the conclusion of your intro. (e.g. "We hypothesized that lemming presence would do X to soil storage…we also hypothesized that lemming presence would reduce both CH4 and CO2 fluxes because of XYZ…lastly, we hypothesized that a year of recovery from lemming herbivory and disturbance would result in XYZ." Right now they blend in with your other text and don't pop out for easy reading of your goals and predictions!*

**2 Materials and methods**

**2.1 Study location**

This study was carried out in Utqiaġvik (formerly Barrow), Alaska (Fig. 1a). Located in the Arctic Coastal Plain, Utqiaġvik is comprised of flat-, low-, and high-center ice-wedge polygons that cover roughly 65% of the land cover (Billings and Peterson,

1980). The major vegetation type at this site is graminoid-dominated wetlands, consisting of mosses, lichens, graminoids, and wet sedges (Davidson et al., 2016).

*this is a little jargon-y for a biogeochemical audience I suspect! wonderful descriptors but i might suggest simplifying the language. (line 84)*

The study area was located near the Barrow Atmospheric Baseline Observatory and an atmospheric monitoring site managed by the National Oceanic and Atmospheric Administration (NOAA) (Fig. 1b), approximately 2 km south of the Arctic Ocean and

1 m elevation above sea level (71°19′21.10″ N: 156°36′33.04″ W). This site was near a pre-established remote flux and

*cool!!*

meteorological tower monitored by the Global Change Research Group (Goodrich et al., 2016) and has substantial lemming populations relative to other Arctic tundra areas in Alaska (Ott and Currier, 2012).

**Figure 1. (a)** The location of the study site, Utqiaġvik (Barrow), in Alaska (© Google Maps 2018, imagery from TerraMetrics) represented by a red star, **(b)** location of the sampling site (© Google Maps 2018, imagery from TerraMetrics), **(c)** distribution of the sampling plots on an image created using the coordinates of the plots in R (Worldview-3 panchromatic imagery taken 24 July 2016, Maxar Technologies), **(d)** chamber used for the greenhouse gas flux measurements, **(e)** chamber covered by light-blocking material, and **(f, g)** enclosures installed at each of the plots during the manipulation experiment.

*would suggest saving this (line 95) for the supplement, or methods if it's important! e.g. "all sampling sites located on preexisting maps from Worldview-3 using R Studio software" or similar.*

**2.2 Brown lemmings as a study species**

[revised manuscript text omitted]

$\pm 0.012$ gC-$CO_2$m$^{-2}$h$^{-1}$ (i.e. $CO_2$ sequestration) to $0.003 \pm 0.012$ gC-$CO_2$m$^{-2}$h$^{-1}$ (i.e. no different from zero).

*move sentence on line 245-246 to after your reference to fig. 3a; feels funky to go from co2, to ch4, back to co2*

In summer 2019, we measured $CH_4$ fluxes and NEE again, and additionally calculated ecosystem respiration (ER) and gross primary production (GPP). During this second summer of measurements, NEE, ER, GPP, and $CH_4$ flux were all not significantly different between control and lemming plots (NEE $P = 0.834$, Fig. 3b; ER $P = 0.742$, Fig. 5a; GPP $P = 0.716$, Fig. 5b; and $CH_4$

flux $P = 0.869$, Fig. 4b).    *you might want to stick to a single defined reference for NEE/CO2 flux; in 2018 results you use CO2 flux, and here you use NEE. (e.g. does CO2 flux include plants, or is the same as ER, etc.) Since these terms \*can\* be fuzzy, I'd suggest defining the variables as you plan to use them in the intro, and then using the same throughout very strictly!*

[Figure]

**Figure 3.** Box and whisker plots of 2018 and 2019 $CO_2$ net ecosystem exchange fluxes for the control and lemming plots. Negative flux values
indicate carbon sequestration/uptake from the atmosphere and positive flux values indicate carbon emission/loss to the atmosphere. (**a**) Median
$CO_2$ flux for plots before and after the experiment in summer 2018 ($T = 4.62$, $P < 0.001$), and (**b**) median $CO_2$ flux for plots during the three
rounds of measurements in summer 2019 ($T = 0.21$, $P = 0.834$).

[Figure]

**Figure 4.** Box and whisker plots of 2018 and 2019 $CH_4$ fluxes for control and lemming plots. Negative flux values indicate uptake from the
atmosphere and positive flux values indicate emission to the atmosphere. (**a**) Median $CH_4$ flux for plots before and after the experiment in summer
($T = 0.01$, $P = 0.989$), and (**b**) median $CH_4$ flux for plots during the three rounds of measurements in summer 2019 ($T = -0.17$, $P = 0.869$).

[Figure]

[Figure]

**Figure 5.** Box and whisker plots of $CO_2$ ecosystem respiration and gross primary production fluxes for control and lemming plots during the three rounds of data collection in summer 2019. (**a**) Median $CO_2$ ecosystem respiration flux ($T = -0.34$, $P = 0.742$), and (**b**) median $CO_2$ gross primary production flux ($T = -0.37$, $P = 0.716$). not totally clear to me how these measurements are different from this plot; from my understanding of your methods, you calculated GPP in 2019? again would suggest a table or similar with terms/definitions as you use them here of all your chamber variables!

[revised manuscript text omitted]

*^ these last two paragraphs, I feel, could be condensed into one that focuses on the spatial AND temporal variability in carbon uptake/emission in the tundra over a year, considering the spatial dependence (e.g. heavy herbivory near burrows) and temporal dependence (peak abundance season) of heavy lemming herbivory. then the rest of this section could be speculating how these variabilities in space and time add up to a net effect on the waterlogged tundra, which is what seems to me the most interesting possible application of this work (e.g. what if lemming population increased due to predator loss or some positive effect of climate change or something). As they stand, these paragraphs contain cool and interesting information but not as much synthesis / new interpretations. It shows up in the conclusions but i think should be introduced here (or else they seem like they come out of left field in conclusion section).*

[revised manuscript text omitted]

other more general notes:

I enjoyed reading this immensely, as I think it's a really cool experimental setup. My comments refer mostly to the framing of the write-up as I think the strength of the experimental design is not currently shining through as well as it could given how cool I personally took it to be! I would suggest that the authors re-frame the paper to exploring the LEGACY effects of heavy lemming herbivory, by quantifying what 'normal' herbivory is in the site and how much more herbivory the experimental enclosures experienced over 16 days (perhaps using the existing NDVI data) to demonstrate the difference. The interpretation and discussion could then probe the circumstances in which such heavy herbivory could occur (e.g. climate-change-induced lengthening of snow-free time; loss of predation; an eruption in lemming population size; etc.) and the resilience of the ecosystem in terms of its recovery after such an event.

important to define the variables you measured/refer, as well as the timeline of your experiment, to very specifically in a table or list (see in-line comments)

what are you referring to when you refer to carbon uptake? I suspect it will be important to define it for reader. and, when you conclude in discussion that lemming herbivory negatively impacted the sites' ability to sequester carbon, do you mean as aboveground biomass/belowground biomass/belowground deposition of sugars/increased soil microbial respiration/ decomposition of root biomass after tops are eaten? Which of these factors is that that is increasing carbon flux, and why do you posit so?

REALLY need to say how soon after lemming removal you measured all your fluxes. that will change your proposed explanation for the change in flux. (see above) (maybe in the table describing sampling timeline that you proposed in the doc itself?)

important that you refer to what they're measuring carefully; as a first-time reader of this work, my impression was that they were measuring the short and longer-term legacy effects of heavy herbivory (bc lemmings are still present here, so they were not removing herbivory; they were imposing then removing heavier herbivory).

It is worth identifying the predators of lemmings since they end up coming up a few times in discussion/conclusion.

lastly: it is worth circling back to the intro/drive in discussion. It's likely going to feel more closed-loop if the intro brings up why knowing the near-term AND legacy impacts of heavy herbivory on carbon cycling in this system. (Changes to populations of predators? Climate change? etc.) E.g. the conclusions do it, but the intro does not hook the reader with 'why should I be excited that you did this experiment with heavy herbivory'.

in closing: really cool experimental design! While I am suggesting some fairly large revisions here (e.g. in framing of the paper bind some more synthesis/interpretation), given the cool design and the very clear results, I am excited to see how it turns out and would gladly read another version.

---

## Author Comment (AC2)

**Responses to the referee's concluding notes at the bottom of the final page:**

**"I enjoyed reading this immensely, as I think it's a really cool experimental setup. My comments refer mostly to the framing of the write-up as I think the strength of the experimental design is not currently shining through as well as it could given how cool I personally took it to be! I would suggest that the authors re-frame the paper to exploring the LEGACY effects of heavy lemming herbivory, by quantifying what 'normal' herbivory is in the site and how much more herbivory the experimental enclosures experienced over 16 days (perhaps using the existing NDVI data) to demonstrate the difference. The interpretation and discussion could then probe the circumstances in which such heavy herbivory could occur (e.g. climate-change-induced lengthening of snow-free time; loss of predation; an eruption in lemming population size; etc.) and the resilience of the ecosystem in terms of its recovery after such an event."**

Since lemming populations are very cyclic in nature and vary with multiple environmental factors, their densities oscillate (Fauteux et al., 2015; Soininen et al., 2017) and make predicting a sort of 'normal' level of herbivory very challenging. Reports on estimated brown lemming density have found their local density to range from 5 to 65 lemmings per hectare (Ott and Currier, 2012; Alaskan Arctic) and about 0 to 9 lemmings per hectare (Fauteux et al., 2015; Canadian Arctic). However, as mentioned in our manuscript, the use of live-trapping as a technique to estimate density for this species may underestimate the actual density size, as brown lemmings are not readily captured using baited traps; we found manual capture techniques to be much more effective than baited traps. Moreover, in addition to space, it is important to consider time: we only kept lemmings inside the plots for 16 hours and there was no effect of lemming herbivory for any of the remaining time. The best comparison we could find to define the degree of herbivory was the observed effect on vegetation near their burrows and runways in published studies from similar ecosystems (Erlinge et al., 2011). Given this, it is extremely difficult to define the difference between 'normal' and 'heavy' herbivory. We agree that separating the effect of normal and heavy legacy effects is very important, but would be extremely difficult with current available data from these understudied Arctic ecosystems. In the revised manuscript, we will include a more careful discussion of this important point, and discuss the issues related to defining the population size and the degrees of herbivory.

**"important to define the variables you measured/refer, as well as the timeline of your experiment, to very specifically in a table or list (see in-line comments)"**

We will specify in more details the types of measurements collected during each summer in the revised manuscript. A summary of these measurements is included below:

|  | **Data Collected** | **Frequency of Measurement** |
|---|---|---|
| **Summer 2018** | $CO_2$ (NEE) and $CH_4$ fluxes, NDVI, air temperature, soil temperature, soil moisture, thaw depth, motion camera footage | Before and after lemming treatment |
| **Summer 2019** | $CO_2$ (NEE, ER, GPP) and $CH_4$ fluxes, NDVI, air temperature, soil temperature, soil moisture, thaw depth | Pre-, early, peak growing season |

**"what are you referring to when you refer to carbon uptake? I suspect it will be important to define it for reader. and, when you conclude in discussion that lemming herbivory negatively impacted the sites' ability to sequester carbon, do you mean as aboveground biomass/belowground biomass/belowground deposition of sugars/increased soil microbial respiration/decomposition of root biomass after tops are eaten? Which of these factors is that that is increasing carbon flux, and why do you posit so?"**

Carbon 'uptake', or 'sequestration', in this context is the removal of $CO_2$ from the atmosphere and its storage in the above- and belowground biomass through photosynthesis. Therefore, when the lemming consumes the photosynthetic tissues of the vegetation (aboveground biomass), the $CO_2$ fluxes increase since the vegetation is no longer able to uptake $CO_2$ from the atmosphere. We will clarify this in the revised manuscript.

**"REALLY need to say how soon after lemming removal you measured all your fluxes. that will change your proposed explanation for the change in flux. (see above) (maybe in the table describing sampling timeline that you proposed in the doc itself?)"**

Lemmings were only placed in our experimental plots during the first summer of measurements (2018); the second summer of measurements (2019) was for measuring the recovery of the vegetation that had been affected by the lemmings during the first summer. In 2018 (the first season of measurements, when we included lemmings inside the manipulation plots) $CO_2$ and $CH_4$ concentrations were measured one day after lemming removal from the plots (exact time varied based on weather conditions and when plots were measured in temporal relation to other plots). We will include these details in the revised manuscript.

**"important that you refer to what they're measuring carefully; as a first-time reader of this work, my impression was that they were measuring the short and longer-term legacy effects of heavy herbivory (bc lemmings are still present here, so they were not removing herbivory; they were imposing then removing heavier herbivory)."**

We will clarify this in the revised manuscript, and include a more careful discussion of the challenges in defining the degree of herbivory.

**"It is worth identifying the predators of lemmings since they end up coming up a few times in discussion/conclusion."**

We specified this on line 7 of the submitted supplemental materials to this manuscript: predators include the snowy owl, parasitic jaeger (arctic skua), arctic fox, and ermine. We will further clarify this in the revised manuscript.

**"lastly: it is worth circling back to the intro/drive in discussion. It's likely going to feel more closed-loop if the intro brings up why knowing the near-term AND legacy impacts of heavy herbivory on carbon cycling in this system. (Changes to populations of predators? Climate change? etc.) E.g. the conclusions do it, but the intro does not hook the reader with 'why should I be excited that you did this experiment with heavy herbivory'."**

We will mention this in the introduction, and make sure the larger implications of our experiment are clear.

**"in closing: really cool experimental design! While I am suggesting some fairly large revisions here (e.g. in framing of the paper and some synthesis/interpretation), given the design and the very clear results, I am excited to see how it turns out and would gladly read another version."**

We thank the referee again for the very helpful comments and suggestions, which will majorly improve the manuscript.

---

## Author Response (AR1)

Global Change Research Group
Department of Biology
San Diego State University
5500 Campanile Dr.
San Diego, CA 92182-4614

Tel: (619) 594-6613
Fax: (619) 594-7831
http://www.sci.sdsu.edu/GCRG/

*31 March 2022*

**Paul Stoy**
*Associate Editor*
*Biogeosciences*
*European Geosciences Union*

*Dear Editor,*

*Please find attached our revised manuscript entitled* **"Response of vegetation and carbon fluxes to brown lemming herbivory in Northern Alaska"** *for possible publication in your journal.*

*We would like to thank you and the referees for the helpful suggestions which substantially improved the manuscript. We include below point-by-point responses to the referees' comments, and detailed descriptions of the modifications we made to the manuscript (with line numbers for the manuscript with track-changes off). We hope that with these changes you will find our revised manuscript appropriate for publication in your journal.*

*Best Regards,*
*Jessica Plein*

*Department of Biology*
*San Diego State University*

**Referee Comment #1:**

**Plein and others explore the response of greenhouse gas fluxes to lemming herbivory on the North Slope of Alaska. Lemmings reduced NDVI and CO2 uptake that recovered the following year due to vegetation regrowth but had little impact on CH4 fluxes.**

**I found the study to be relevant and interesting to the readership of Biogeosciences; studies of the importance of herbivores on GHG fluxes are too infrequent and careful analyses like this help us appreciate the role of biology in the earth system. At the same time I was surprised at the idea that herbivory would decrease CH4 fluxes. Is there a wounding response in the aerenchyma that blocks their role as a CH4 conduit? All else being equal I would assume that herbivory might slightly increase CH4 fluxes that would be very difficult to see from a signal-to-noise perspective because the path that CH4 has to take to the atmosphere has been decreased slightly (by shorter aerenchyma, this won't be much of an effect) or possibly increased if herbivory was enough to reduce transpiration to the point that it impacts groundwater levels. It's hard to see how lemming herbivory would be sufficient to do this outside of massive lemming herbivory of which I am unaware, but if sedges simply regrow after herbivory there shouldn't be much of a methane impact, which was found. The idea that it escapes from the stem bases may hold if the stems help create preferential flow pathways in the soil, but this probably wouldn't compensate for any changes to the aerenchyma. Basically there should be no observable effect on CH4 which is what was found. Interesting to study nonetheless. I had few other concerns and feel that the manuscript is publishable with minor revisions after finding more literature basis for the impacts of herbivory on aerenchyma and subsequent expectations for CH4 flux.**

We thank the referee for the useful comments, which we attempted to address in the revised manuscript. We modified the wording of the hypothesis at the end of the Introduction section and included more references. We more thoroughly explained how the removal of vegetation could affect $CH_4$ emissions, given that vegetation contributes not only to $CH_4$ transport from the soil to the atmosphere, but also the release of labile carbon in the soil, which stimulates $CH_4$ emission (lines 35-40). In the Discussion section, we elaborate on why the $CH_4$ results may not be what we expected due to how lemming herbivory affected the vegetation through the location of clipping and photosynthetic activity interacting with labile carbon (lines 343-358). Additionally, we discuss how lemming urine may counter the impact of changes to the aerenchyma on $CH_4$ emission, as ammonium from urine has been linked to an increase in $CH_4$ production, but may also result in a mean negative flux shortly after the addition of urine to the system (lines 359-365).

**Referee Comment #2:**

**I have included in-line notes I took while reading through the manuscript for the first time, and some concluding notes (at the bottom of the final page) on my interpretation of the entire manuscript. As noted there, while I am suggesting major revisions I am excited by the experimental design and data of this paper and would gladly look at it again.**

We thank the referee for the positive evaluation of our study and for the useful comments, which we incorporated in the revised manuscript. Below, we included answers to the comments listed in the referee's concluding notes.

**Here are the concluding notes at the bottom of the final page:**

**I enjoyed reading this immensely, as I think it's a really cool experimental setup. My comments refer mostly to the framing of the write-up as I think the strength of the experimental design is not currently shining through as well as it could given how cool I personally took it to be!  I would suggest that the authors re-frame the paper to exploring the LEGACY effects of heavy lemming herbivory, by quantifying what 'normal' herbivory is in the site and how much more herbivory the experimental enclosures experienced over 16 days (perhaps using the existing NDVI data) to demonstrate the difference.  The interpretation and discussion could then probe the circumstances in which such heavy herbivory could occur (e.g. climate-change-induced lengthening of snow-free time; loss of predation; an eruption in lemming population size; etc.) and the resilience of the ecosystem in terms of its recovery after such an event.**

This is a very important point which we tried to address as best as we could with the data available in the literature. In the Discussion section (lines 390-399), we discuss that since lemming population densities vary in response to multiple environmental factors (Fauteux et al., 2015; Soininen et al., 2017), predicting a 'normal' level of herbivory for this species is very challenging. Reports on estimated brown lemming density have found their local density to range from five to 65 lemmings per hectare (Ott and Currier, 2012; Alaskan Arctic) and about zero to nine lemmings per hectare (Fauteux et al., 2015; Canadian Arctic). However, as mentioned in the Materials and Methods section of the revised manuscript (lines 134-136), the use of live-trapping as a technique to estimate density for this species may underestimate the actual population density. Moreover, in addition to space, it is important to consider time: we only kept lemmings inside the plots for 16 hours and there was no effect of lemming herbivory for the remainder of the experiment. The scientific literature on lemming herbivory is extremely sparse, and the most relevant comparison we could find to define the degree of herbivory observed was the effect on vegetation near lemming burrows and runways in a similar ecosystem (Erlinge et al., 2011; Siberian tundra). We agree that separating the effect of normal and heavy legacy effects is very important, but given the sparsity of available literature and data from these understudied Arctic ecosystems, it is difficult to categorize our lemming treatment as having some sort of 'normal' or 'heavy' impact on vegetation, which would be required to explore legacy effects of lemming herbivory.

**important to define the variables you measured/refer, as well as the timeline of your experiment, to very specifically in a table or list (see in-line comments)**

We added a table listing the data collected and frequency of measurement during both field seasons in the Materials and Methods section (see Table 1 in the revised manuscript), with the terms in the caption of Table 1 (lines 179-182). We also added more precise dates of data collection (lines 140, 177-178).

**what are you referring to when you refer to carbon uptake?  I suspect it will be important to define it for reader.**

We included more details of the different terms we used in the Materials and Methods section. Specifically, we mentioned that the net ecosystem exchange (NEE) used in equation 1 (and shown in Fig. 2) was the net balance between the carbon uptake from photosynthesis and the carbon loss from respiration (lines 202-205). In the Results section, we further elaborated on the definition of carbon uptake in the context of this study (lines 274-275). We also specified in lines 301-302 of the revised manuscript (caption of Fig. 4) the meaning of carbon uptake and explained the meaning of the signs for NEE, GPP, and ER in Fig. 2 and Fig. 4 captions.

**and, when you conclude in discussion that lemming herbivory negatively impacted the sites' ability to sequester carbon, do you mean as aboveground biomass/belowground biomass/belowground deposition of sugars/increased soil microbial respiration/decomposition of root biomass after tops are eaten? Which of these factors is that that is increasing carbon flux, and why do you posit so?**

This is a very important point that should be investigated in future studies. Unfortunately, the design of this experiment, mostly focusing on the aboveground measurements (except for the soil temperature, soil moisture, and thaw depth), did not allow for identifying the contribution of belowground increased decomposition from the aboveground vegetation removal. However, as we did not notice an increase in $CH_4$ emission with vegetation removal (which could have increased as an indirect effect of an increase in sugars related to increased soil microbial respiration), we could assume that the direct effect of the removal of photosynthetic plant tissue was the main mechanism explaining the decrease in the ability of the ecosystem to sequester carbon. We included this in the Discussion section (lines 335-342).

**REALLY need to say how soon after lemming removal you measured all your fluxes. that will change your proposed explanation for the change in flux. (see above) (maybe in the table describing sampling timeline that you proposed in the doc itself?)**

We included this detail in the Materials and Methods section (lines 192-194). The fluxes were measured one day after the removal of the lemming from each plot in summer 2018 (when we performed a manipulative experiment).

**important that you refer to what they're measuring carefully; as a first-time reader of this work, my impression was that they were measuring the short and longer-term legacy effects of heavy herbivory (bc lemmings are still present here, so they were not removing herbivory; they were imposing then removing heavier herbivory).**

We included these details in lines 64-70 of the Introduction section, and also added Table 1, which includes a full list of all the data collected during each of the field seasons and the frequency of these measurements (2018: before and after lemming treatment; 2019: pre-, early, and peak growing season).

**It is worth identifying the predators of lemmings since they end up coming up a few times in discussion/conclusion.**

We identified the predators of lemmings in the Materials and Methods section (line 119) of the revised manuscript.

**lastly: it is worth circling back to the intro/drive in discussion. It's likely going to feel more closed-loop if the intro brings up why knowing the near-term AND legacy impacts of heavy herbivory on carbon cycling in this system. (Changes to populations of predators? Climate change? etc.) E.g. the conclusions do it, but the intro does not hook the reader with 'why should I be excited that you did this experiment with heavy herbivory'.**

We added more details of the potential relevance and larger implications of our study in lines 81-84 of the Introduction section. We feel that this study may help increase our understanding of how herbivores impact carbon fluxes and the photosynthetic capacity of plants in the Alaskan Arctic environment, and hope to further interest in complex interactions in the Arctic.

**in closing: really cool experimental design! While I am suggesting some fairly large revisions here (e.g. in framing of the paper and some synthesis/interpretation), given the design and the very clear results, I am excited to see how it turns out and would gladly read another version.**

We thank the referee for the suggestions and hope the revisions are found to have strengthened the manuscript.

**Referee Comment #3:**

**The manuscript, entitled "Response of vegetation and carbon fluxes to brown lemming herbivory in Northern Alaska" by Jessica Plein and coauthors (bg-2021-286) proposes in a research paper to study how brown lemming impacted the short-term response and the recovery of vegetation carbon uptake, NDVI and CO2 and CH4 emission to atmosphere. The study shows that the NDVI and carbon uptake by vegetation was immediately impacted by lemming placed in the enclosure, while there was no impact during the next-year recovery for every variables of the tundra functioning.**

**This manuscript is written with a perfect English and the experimentation has been conducted thoroughly. While only a few studies focus on small herbivore impact on C fluxes in Arctic, I found that the study is not extensive / comprehensive enough to be proposed in a large audience journal like Biogeoscience. Fundamentally, the study focuses on vegetation disturbance, which is due to the introduction of lemming enclosure but it could be any other source of disturbance and the response would have been the same. As such, the response shows nothing specific to lemming. Plant community is totally lacking at the study and the interpretation. This is essential as lemming is quite specific in its plant species diet and CH4 efflux can be promoted by species characterised by aerenchyma. As such, this could be a way to show the specificity of lemming herbivory. Of secondary importance, there are several parts I found too long and misplaced. Below I present more precisely the different parts that could be improved:**

We thank the referee for their detailed recommendations, which we incorporated into our revised manuscript. We included in our revisions that we believe the main source of disturbance resulting in removal of vascular plants in these wet tundra ecosystems to be from lemming herbivory. There could be other sources of herbivory (such as caribou), but they are not as frequent in these northernmost areas of the Arctic Coastal Plain. Additional sources of disturbance to vegetation could originate from a drastic change in environmental conditions, such as extreme temperatures, extremely dry conditions, etc.; however, these would not selectively remove the vascular plants while not affecting the moss layer, which is what we observed in this experiment (lines 184-187). We measured a wide range of environmental variables in both control and experimental plots, as we described in lines 264-265, which showed no difference. Moreover, we described the vegetation community in the Materials and Methods section (lines 89-90), and added the vegetation types (mosses, lichens, graminoids, and wet sedges) into the sampling plan and experimental design subsection to emphasize the vegetation in our plots (lines 148-149). When designing our experiment (lines 144-146), we took into consideration the preferential diet of lemmings in summer (lines 110, 368-371). In the Discussion section, we mentioned that we used motion-sensor camera footage to observe that lemming foraging within the plots was representative of their vegetation preferences, and that those were the types of vegetation removed by the lemmings (lines 371-372). We also noted in the Materials and Methods and Discussion sections (lines 150-151, 372-373) that an in-depth analysis of the vegetation types found in our plots was completed in a previous study by our team (Davidson et al., 2016).

**L10 and L25: This is a repetition and I propose to remove the one of the abstract**

We removed this statistic from the Abstract section of the revised manuscript.

**L182: GPP is the annual flux, while only three instantaneous measurements were recorded. It would be more accurate to use a name that better describes the variable.**

Since plant growth and photosynthetic uptake is restricted to the summer months in these Arctic ecosystems, we used GPP to indicate "the total amount of $CO_2$ 'fixed' by land plants per unit time through the photosynthetic reduction of $CO_2$ into organic compounds" (Gough, 2011) during the time of measurements, rather than as an annual measurement. We clarified this when introducing GPP in the Materials and Methods section (lines 209-212).

**L208ff: Can you give details on the distribution of data and the one used in the regression?**

We included this information on the data distribution in the Materials and Methods section of the revised manuscript (lines 239-247). We tested for normality using a Shapiro-Wilk normality test. The 2018 data were normally distributed (NEE $P = 0.489$, NDVI $P = 0.816$), except the $CH_4$ data ($P < 0.001$), which were right-skewed, so we log-transformed the $CH_4$ data to help normalize them ($P = 0.284$). After this transformation, we used linear mixed-effects models to test for the significance between the different treatments. For the 2019 data, we used linear mixed-effects models (like we did in 2018) and non-parametric Kruskal–Wallis tests because we could not make all the data normal using the same transformation method (log transformation, square root transformation) for every round during the season. Results of the Kruskal–Wallis tests were consistent with those of the linear mixed-effects models (see Results section; lines 282-289, 317-321). We also tested for equal variance using an F-test and found there was no significant difference between the variances (treatments) in 2018 (NEE $P = 0.172$, $CH_4$ flux $P = 0.810$, NDVI $P = 0.100$) and 2019 (NEE $P = 0.441$, ER $P = 0.650$, GPP $P = 0.852$, $CH_4$ flux $P = 0.346$, NDVI $P = 0.951$). We plotted the models to examine the residuals of the data and found them to not appear heteroscedastic.

**Figure 2 should be placed in the appendix or a more concise presentation should be drawn.**

We moved this figure (now Fig. A1) to the Appendix A section in the revised manuscript.

**L239: It would be important to give the atmospheric concentration of CH4.**

We included the global mean atmospheric concentrations of both $CO_2$ and $CH_4$ when mentioning greenhouse gas concentrations in the Materials and Methods section (lines 190-191).

**Figure 3 shows primary productivity with negative values, which is totally comprehensive. However, figure 5 shows the same variable with positive value, which is both not comprehensive and standardised with Fig. 3.**

We more clearly explained the signs of the flux values in the captions of the figures in the revised manuscript. We mentioned that figure 3 (now Fig. 2) shows the net ecosystem exchange of 2018 and 2019, with negative values indicating the removal of $CO_2$ from the atmosphere by vegetation through photosynthesis, and positive values indicating the carbon loss into the atmosphere (lines 291-293). We also mentioned that figure 5 (now Fig. 4) shows the ecosystem respiration (4a) and gross primary production (4b), but in this case the positive sign is indicating a positive respiration (carbon loss into the atmosphere) and a positive carbon uptake by vegetation

through photosynthesis (lines 301-302). We included that the signs of ER and GPP are always positive, but if ER is more than GPP, then the ecosystem is a carbon source into the atmosphere (with a positive sign of NEE) (lines 302-303). In the Materials and Methods section, we noted our usage of the equation GPP = NEE – ER and the sign convention suggested by Chapin et al. (2006) (lines 208-209).

**L324ff: This paragraph seems to belong to introduction or material and method but not to the discussion.**

We moved most of this paragraph to the Materials and Methods section, where the information is more relevant (lines 119-125). We saved part of the paragraph in the Discussion section for suggested areas of future research (lines 400-402).

**I read your paper with great interest and I belief it is relevant to arctic ecology readership, providing the consideration of the issues presented here, especially the inclusion of vegetation species and traits. The long-term impact will be very much interesting and I encourage the authors to continue this much valuable and important work.**

We thank the referee for their useful suggestions and hope they find our revised manuscript appropriate for the Biogeosciences audience.

---

## Referee Report (RR1)

47: Few rodent species persist throughout the Arctic. Lemmings are by far the most abundant and widespread and are consequently identified as a keystone species in tundra environments (Krebs 2011). *Current wording contains confusing negatives.*

65: Somewhat awkward phrasing that makes the reader search for what the 'gap' in question is (carbon cycling). Could consider rephrasing this sentence to be more clear, e.g. "The current body of literature does not explore lemming impacts on carbon cycling, leaving a crucial gap in our understanding of how one of the main herbivores influences this rapidly changing ecosystem especially in light of Arctic warming."

70: would suggest changing "the disturbance" to "herbivory" or "targeted herbivory" because the authors suggest in their response to reviewers that the level of herbivory experienced in the enclosures is normal. "Disturbance" suggests a treatment effect that is above normal or what the enclosed areas would experience without the experiment.

76: remove "their" – currently it implies the lemmings are recovering, not the vegetation

79: we measured what in the plots? Suggest: "…, we measured vegetation in the plots again to evaluate vegetation recovery from grazing"

88-92: some issues with tense here, as previously the authors used exclusively past tense. Suggest changing here or committing to present tense elsewhere, and double-checking for consistency throughout. Additionally, line 88 seems to be leftover from a previous draft? I think this part is meant to include only line 89 on.

104: I still really love this figure. Too few of these kinds of papers include some vivid photographs of the experimental setup, and it's so helpful!

132: this seems like an example of the previous sentence, so "additionally" feels like an awkward word choice. Perhaps, "to wit,", or "for example,", etc.?

149-152: repeated paragraph? Delete

168: suggest "for inclusion in the experiment". Current phrasing is a bit confusing, implies the lemmings were immediately placed in the enclosures.

172-173: confusing phrasing. Suggest: "Each plot contained different vegetation types (mosses, lichens, graminoids, and wet sedges) and each pair was ensured to be as similar in composition as possible…"

175-176: order of phrasing here is confusing. Suggest merging these two sentences. "We placed control plots within 1m of their paired lemming plot to keep environmental factors as similar as possible within pairs; we located each pair of plots approximately 3m away from each other."

206: this table is great!  Really makes the sampling timeline clear, and what the authors measured.  Very much appreciate this addition and think it adds a lot.

232: would be helpful to use consistent phrasing to describe each summer.  "following" and "subsequent" seem like two different summers despite both being 2019.  I suggest using "…before and after the the first summer's manipulations (2018) as previously described and to track the seasonal development of NEE during the second summer (2019). In the second summer, we used a …"

406, 407: remove these lines (repeats the following line)

423-427: this paragraph seems like it should be placed in the same paragraph as the one above, preceding that text (e.g. placed at line 415 before the text that is currently there).  In its current placement the information feels redundant even though the authors are discussing different data.

452-456: these repeated lines seem like a formatting error but highlighting in any case

457-461: are the authors referring to changes to spatial effects on vegetation given the risk of predation and its impacts on lemming foraging behavior?  I think this final paragraph could be fleshed out a tiny bit more; what kinds of changes to foraging behavior are they referring to and how would that change carbon storage/cycling?  My first instinct is spatial changes in carbon cycling given changes to lemming foraging, but risk of predation could also cause more physiological changes to lemming populations that would indirectly affect carbon (see Hawlena and Schmitz work on fear effects on grasshopper behavior and physiology).  But the long and short of it is that this final paragraph could use 1-2 specific examples of the kinds of proposed predator effects on carbon cycling re: effects on lemmings that the authors are discussing here (especially as they bring up predators in their conclusions, which otherwise don't occur in the body of the ms).

475: replace "the" with "carbon"

TAKEAWAY:
The reading of the manuscript has improved greatly with the inclusion of the reviewers' comments.  I appreciate the efforts the authors have made and the changes asked for, and think that the final product is clearer, more "punchy", and that the substantial research summarized therein is highlighted much more effectively.  The above line edits are, for the most part, minor, with the exception of my one comment on adding a few sentences to the discussion (lines 457-461, highlighted in green for ease).  I commend the authors on the work they have done, and am happy to recommend the paper to be accepted with technical corrections and handed over to the journal's editors for that work with the authors.

---

## Author Response (AR2)

**Global Change Research Group**
**Department of Biology**
**San Diego State University**
**5500 Campanile Dr.**
**San Diego, CA 92182-4614**

**Tel:** **(619) 594-6613**
**Fax:** **(619) 594-7831**
**http://www.sci.sdsu.edu/GCRG/**

*27 April 2022*

*Dr. Paul C. Stoy*
*Associate Editor*
*Biogeosciences*
*European Geosciences Union*

*Dear Editor,*

 *Please find attached our revised manuscript entitled* **"Response of vegetation and carbon fluxes to brown lemming herbivory in Northern Alaska"** *for publication in your journal.*

 *We would like to thank you and the referees again for the great feedback. We include below point-by-point responses to the referees' technical corrections (with line numbers for the manuscript with track-changes off). We hope that with these corrections you will find our manuscript appropriate for publication in your journal.*

*Best Regards,*
*Jessica Plein*

*Department of Biology*
*San Diego State University*

**Referee Technical Corrections:**

**47: Few rodent species persist throughout the Arctic. Lemmings are by far the most abundant and widespread and are consequently identified as a keystone species in tundra environments (Krebs 2011).** *Current wording contains confusing negatives.*

We changed this wording to: "Throughout the Arctic, lemmings are by far the most abundant and widespread rodent species, and are identified as keystone species in tundra environments (Krebs, 2011)" (lines 43-44).

**65: Somewhat awkward phrasing that makes the reader search for what the 'gap' in question is (carbon cycling). Could consider rephrasing this sentence to be more clear, e.g. "The current body of literature does not explore lemming impacts on carbon cycling, leaving a crucial gap in our understanding of how one of the main herbivores influences this rapidly changing ecosystem especially in light of Arctic warming."**

We changed the phrasing to: "The current body of literature does not explore the direct impact of lemming presence on carbon cycling and vegetation recovery, leaving a crucial gap in our understanding of how one of the main herbivores influences this rapidly changing ecosystem, especially in light of Arctic warming" (lines 59-61).

**70: would suggest changing "the disturbance" to "herbivory" or "targeted herbivory" because the authors suggest in their response to reviewers that the level of herbivory experienced in the enclosures is normal. "Disturbance" suggests a treatment effect that is above normal or what the enclosed areas would experience without the experiment.**

We changed this wording, as suggested (lines 65, 67).

**76: remove "their" – currently it implies the lemmings are recovering, not the vegetation**

We removed "their" (line 71).

**79: we measured what in the plots? Suggest: "…, we measured vegetation in the plots again to evaluate vegetation recovery from grazing"**

We specified that vegetation was measured (line 74).

**88-92: some issues with tense here, as previously the authors used exclusively past tense. Suggest changing here or committing to present tense elsewhere, and double-checking for consistency throughout. Additionally, line 88 seems to be leftover from a previous draft? I think this part is meant to include only line 89 on.**

The tense was changed to past tense for consistency with the rest of the manuscript (lines 81-84). The line 88 referenced by the referee was a line we removed but was not showing as removed in the PDF version including the track-changes.

**104: I still really love this figure. Too few of these kinds of papers include some vivid photographs of the experimental setup, and it's so helpful!**

Thank you for your feedback on this! We want readers to be able to easily visualize the experimental setup.

**132: this seems like an example of the previous sentence, so "additionally" feels like an awkward word choice. Perhaps, "to wit,", or "for example,", etc.?**

We changed "additionally" to "to wit" (line 113).

**149-152: repeated paragraph? Delete**

This paragraph does not repeat in the version with track-changes off, but was not showing as such in the PDF version including the track-changes.

**168: suggest "for inclusion in the experiment". Current phrasing is a bit confusing, implies the lemmings were immediately placed in the enclosures.**

We changed this phrasing, as suggested (line 144).

**172-173: confusing phrasing. Suggest: "Each plot contained different vegetation types (mosses, lichens, graminoids, and wet sedges) and each pair was ensured to be as similar in composition as possible…"**

We modified the phrasing to improve clarity, as suggested (lines 148-150).

**175-176: order of phrasing here is confusing. Suggest merging these two sentences. "We placed control plots within 1m of their paired lemming plot to keep environmental factors as similar as possible within pairs; we located each pair of plots approximately 3m away from each other."**

We merged the two sentences to improve clarity, as suggested (lines 151-153).

**206: this table is great! Really makes the sampling timeline clear, and what the authors measured. Very much appreciate this addition and think it adds a lot.**

We agree that this table was a great suggestion and addition to the manuscript!

**232: would be helpful to use consistent phrasing to describe each summer. "following" and "subsequent" seem like two different summers despite both being 2019. I suggest using "…before and after the the first summer's manipulations (2018) as previously described and to track the seasonal development of NEE during the second summer (2019). In the second summer, we used a …"**

We modified the phrasing, as suggested by the referee (lines 204-206).

**406, 407: remove these lines (repeats the following line)**

These lines do not repeat in the version with track-changes off, but was not showing as such in the PDF version including the track-changes.

**423-427: this paragraph seems like it should be placed in the same paragraph as the one above, preceding that text (e.g. placed at line 415 before the text that is currently there). In its current placement the information feels redundant even though the authors are discussing different data.**

We moved the paragraph to the location recommended by the referee (lines 361-366).

**452-456: these repeated lines seem like a formatting error but highlighting in any case**

Yes, it appears the referee is correct about this being an error. These lines do not repeat in the version with track-changes off, but was not showing as such in the PDF version including the track-changes.

**457-461: are the authors referring to changes to spatial effects on vegetation given the risk of predation and its impacts on lemming foraging behavior? I think this final paragraph could be fleshed out a tiny bit more; what kinds of changes to foraging behavior are they referring to and how would that change carbon storage/cycling? My first instinct is spatial changes in carbon cycling given changes to lemming foraging, but risk of predation could also cause more physiological changes to lemming populations that would indirectly affect carbon (see Hawlena and Schmitz work on fear effects on grasshopper behavior and physiology). But the long and short of it is that this final paragraph could use 1-2 specific examples of the kinds of proposed predator effects on carbon cycling re: effects on lemmings that the authors are discussing here (especially as they bring up predators in their conclusions, which otherwise don't occur in the body of the ms).**

In the revised manuscript, we discussed in more details how we expect predators to affect the tundra carbon cycling (lines 394-403). Lemming populations may vary in response to regulation by predators (Fauteux et al., 2018b), and predation risk may change lemming physiological response and foraging behavior (Hawlena and Schmitz, 2010). In many terrestrial systems, indirect effects of predator presence on herbivores have been shown to have dramatic effects on vegetation consumption (Apfelbach et al., 2005; Borowski, 1998), with resulting behavioral changes rippling through the ecosystem (Ripple and Beschta, 2003). Given the substantial impact of lemming herbivory on the tundra carbon balance, indirect cues indicating predator presence may alter lemming behavior and thus vegetation. If predator cues elicit a fear response in the lemmings, therefore decreasing the time spent consuming vegetation, this change in behavior may decrease the severity of lemmings' impact on vegetation and carbon cycling, specifically their negative affect on $CO_2$ sequestration.
In addition to the experiment outlined in our manuscript, we also incorporated predator cues from major predators of lemmings in order to examine the potential for indirect effects of predators on herbivory rates (see supplementary materials). However, the sample size of our predator manipulation was too small given some logistical issues related to the data collection for this aspect of the experiment, so we suggest future studies to better quantify the influence of predator-prey interactions on herbivory, and how they further impact vegetation and carbon fluxes in the Arctic tundra.

**475: replace "the" with "carbon"**

We replaced the word choice, as recommended (line 417).

**TAKEAWAY:**
**The reading of the manuscript has improved greatly with the inclusion of the reviewers' comments. I appreciate the efforts the authors have made and the changes asked for, and think that the final product is clearer, more "punchy", and that the substantial research summarized therein is highlighted much more effectively. The above line edits are, for the most part, minor, with the exception of my one comment on adding a few sentences to the discussion (lines 457-461, highlighted in green for ease). I commend the**

**authors on the work they have done, and am happy to recommend the paper to be accepted with technical corrections and handed over to the journal's editors for that work with the authors.**

We thank the referee for their very helpful comments and detailed suggestions. We agree the manuscript is much stronger with the inclusion of the referee feedback, and appreciate the recommendation of the paper.